# COMPLETING VISUAL OBJECTS VIA BRIDGING GENERATION AND SEGMENTATION

## ABSTRACT

This paper presents a novel approach to object completion, with the primary goal of reconstructing a complete object from its partially visible components. Our method, named MaskComp, delineates the completion process through iterative stages of generation and segmentation. In each iteration, the object mask is provided as an additional condition to boost image generation, and, in return, the generated images can lead to a more accurate mask by fusing the segmentation of images. We demonstrate that the combination of one generation and one segmentation stage effectively functions as a mask denoiser. Through alternation between the generation and segmentation stages, the partial object mask is progressively refined, providing precise shape guidance and yielding superior object completion results. Our experiments demonstrate the superiority of MaskComp over existing approaches, e.g., ControlNet and Stable Diffusion, establishing it as an effective solution for object completion.

## 1 INTRODUCTION

In recent years, creative image editing has attracted substantial attention and seen significant advancements. Recent breakthroughs in image generation techniques have delivered impressive results across various image editing tasks, including image inpainting (Xie et al., 2023), composition (Yang et al., 2023a) and colorization (Chang et al., 2023). However, another intriguing challenge lies in the domain of object completion. This task involves the restoration of partially occluded objects within an image. Unlike other conditional generation tasks, e.g., image inpainting, which only generates and integrates complete objects into images, object completion requires a seamless alignment between the generated content and the given partial object, which imposes more challenges to recover realistic and comprehensive object shapes.

To guide the generative model in producing images according to a specific shape, additional conditions can be incorporated (Koley et al., 2023; Yang et al., 2023b). Image segmentation has been shown to be a critical technique for enhancing the realism and stability of generative models by

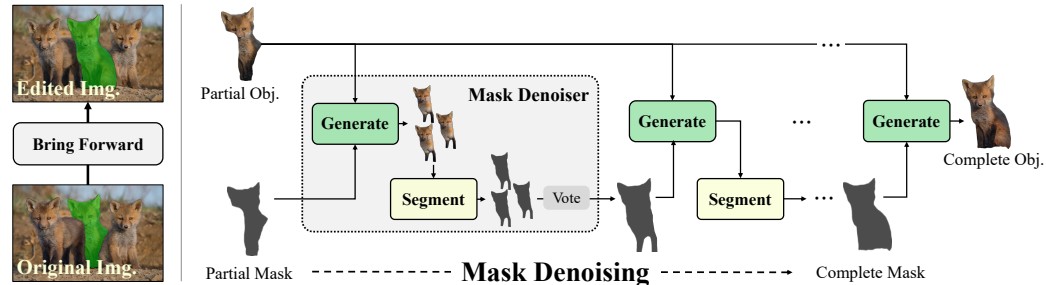

Figure 1: **Illustration of iterative mask denoising (IMD).** Starting from an initial partial object and its corresponding mask, IMD utilizes alternating generation and segmentation stages to progressively refine the partial mask until it converges to the complete mask. With the complete mask as the condition, the final complete object can be seamlessly generated.

providing pixel-level guidance during the synthesis process. Recent research, as exemplified in the latest study by Zhang et al. (Zhang et al., 2023), showcases that, by supplying object segmentations as additional conditions for shaping the objects, it becomes possible to generate complex images of remarkable fidelity.

In this paper, we present MaskComp, a novel approach that bridges image generation and segmentation for effective object completion. MaskComp is rooted in a fundamental observation: the quality of the resulting image in the mask-conditioned generation is directly influenced by the quality of the conditioned mask (Zhang et al., 2023). That says the more detailed the conditioned mask, the more realistic the generated image. Based on this observation, unlike prior object completion methods that solely rely on partially visible objects for generating complete objects, MaskComp introduces an additional mask condition combined with an interactive mask denoising (IMD) process, progressively refining the incomplete mask to provide comprehensive shape guidance to the object completion.

Our approach formulates the partial mask as a noisy form of the complete mask and the IMD process is designed to iteratively denoise this noisy partial mask, eventually leading to the attainment of the complete mask. As illustrated in Figure 1, each IMD step comprises two crucial stages: generation and segmentation. The generation stage's objective is to produce complete object images conditioning on the visible portion of the target object and an object mask. Meanwhile, the segmentation stage is geared towards segmenting the object mask within the generated images and aggregating these segmented masks to obtain a superior mask that serves as the condition for the subsequent IMD step. By seamlessly integrating the generation and segmentation stages, we demonstrate that each IMD step effectively operates as a mask-denoising mechanism, taking a partially observed mask as input and yielding a progressively more complete mask as output. Consequently, through this iterative mask denoising process, the originally incomplete mask evolves into a satisfactory complete object mask, enabling the generation of complete objects guided by this refined mask.

The effectiveness of MaskComp is demonstrated by its capacity to address scenarios involving heavily occluded objects and its ability to generate realistic object representations through the utilization of mask guidance. In contrast to recent progress in the field of image generation research, our contributions can be succinctly outlined as follows:

- We explore and unveil the benefits of incorporating object masks into the object completion task. A novel approach, MaskComp, is proposed to seamlessly bridge the generation and segmentation.

- We formulate the partial mask as a form of noisy complete mask and introduce an iterative mask denoising (IMD) process, consisting of alternating generation and segmentation stages, to refine the object mask and thus improve the object completion.

- We conduct extensive experiments for analysis and comparison, the results of which indicate the superiority and robustness of MaskComp against previous methods, e.g., Stable Diffusion.

## 2 RELATED WORKS

### 2.1 CONDITIONAL IMAGE GENERATION

Conditional image generation Van den Oord et al. (2016); Lee et al. (2022); Gafni et al. (2022); Li et al. (2023b) involves the process of creating images based on specific conditions. These conditions can take various forms, such as layout (Li et al., 2020; Sun & Wu, 2019; Zhao et al., 2019), sketch (Koley et al., 2023), or semantic masks (Gu et al., 2019). For instance, Cascaded Diffusion Models (Ho et al., 2022) utilize ImageNet class labels as conditions, employing a two-stage pipeline of multiple diffusion models to generate high-resolution images. Meanwhile, in the work by (Sehwag et al., 2022), diffusion models are guided to produce novel images from low-density regions within the data manifold. Another noteworthy approach is CLIP (Radford et al., 2021), which has gained widespread adoption in guiding image generation in GANs using text prompts (Galatolo et al., 2021; Gal et al., 2022; Zhou et al., 2021b). In the realm of diffusion models, Semantic Diffusion Guidance (Liu et al., 2023) explores a unified framework for diffusion-based image generation with language, image, or multi-modal conditions. Dhariwal et al. (Dhariwal & Nichol, 2021) employ an ablated diffusion model that utilizes the gradients of a classifier to guide the diffusion process, balancing

diversity and fidelity. Furthermore, Ho et al. (Ho & Salimans, 2022) introduce classifier-free guidance in conditional diffusion models, incorporating score estimates from both a conditional diffusion model and a jointly trained unconditional diffusion model.

## 2.2 IMAGE SEGMENTATION

In the realm of image segmentation, traditional approaches have traditionally leaned on domain-specific network architectures to tackle various segmentation tasks, including semantic, instance, and panoptic segmentation (Long et al., 2015; Chen et al., 2015; He et al., 2017; Neven et al., 2019; Newell et al., 2017; Wang et al., 2020b; Cheng et al., 2020; Wang et al., 2021; 2020a; Li et al., 2023a). However, recent strides in transformer-based methodologies, have highlighted the effectiveness of treating these tasks as mask classification challenges (Cheng et al., 2021; Zhang et al., 2021; Cheng et al., 2022; Carion et al., 2020). MaskFormer (Cheng et al., 2021) and its enhanced variant (Cheng et al., 2022) have introduced transformer-based architectures, coupling each mask prediction with a learnable query. Unlike prior techniques that learn semantic labels at the pixel level, they directly link semantic labels with mask predictions through query-based prediction. Notably, the Segment Anything Model (SAM) (Kirillov et al., 2023) represents a cutting-edge segmentation model that accommodates diverse visual and textual cues for zero-shot object segmentation. Similarly, SEEM (Zou et al., 2023) is another universal segmentation model that extends its capabilities to include object referencing through audio and scribble inputs. By leveraging those foundation segmentation models, e.g., SAM and SEEM, a number of downstream tasks can be boosted (Ma & Wang, 2023; Cen et al., 2023; Yu et al., 2023).

## 3 OBJECT COMPLETION VIA ITERATIVE MASK DENOISING

**Problem definition.** We address the task of object completion task, wherein the objective is to predict the image of a complete object $I_c \in \mathbb{R}^{3 \times H \times W}$, based on its visible (non-occluded) part $I_p \in \mathbb{R}^{3 \times H \times W}$.

We first discuss the high-level idea of the proposed **I**terative **M**ask **D**enoising (IMD) and then illustrate the module details in Section 3.1 and Section 3.2. The core of IMD is based on an essential observation: In the mask-conditioned generation, the quality of the generated object is intricately tied to the quality of the conditioned mask. As shown in Fig. 2, we visualize the completion result of the same partial object but

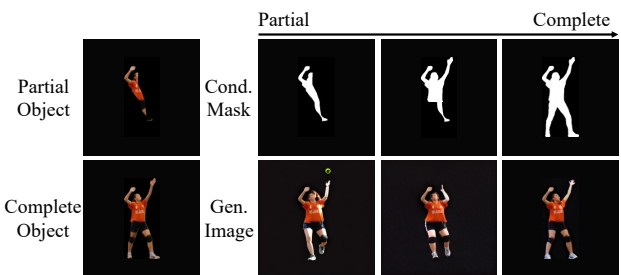

Figure 2: Object completion with different mask conditions.

with different conditioning masks. We notice a more complete object mask condition will result in a more complete and realistic object image. Based on this observation, high-quality occluded object completion can be achieved by providing a complete object mask as the condition.

However, in real-world scenarios, the complete object mask is not available. To address this problem, we propose the IMD process which leverages intertwined generation and segmentation processes to gradually approach the partial mask to the complete mask. Given a partially visible object $I_p$ and its corresponding partial mask $M_p$, the conventional object completion task aims to find a generative model $\mathcal{G}$ such that $I_c \leftarrow \mathcal{G}(I_p)$, where $I_c$ is the complete object. Here, we additionally add the partial mask $M_p$ to the condition $I_c \leftarrow \mathcal{G}(I_p, M_p)$, where $M_p$ can be assumed as an addition of the complete mask and a noise $M_p = M_c + \Delta$. By introducing a segmentation model $\mathcal{S}$, we can find a mask denoiser $\mathcal{S}(\mathcal{G}(\cdot))$ from the object completion model:

$$M_c \leftarrow \mathcal{S}(\mathcal{G}(I_p, M_c + \Delta)) \tag{1}$$

where $M_c = \mathcal{S}(I_c)$. Starting from the visible mask $M_0 = M_p$, as shown in Fig. 1, we repeatedly apply the mask denoiser $\mathcal{S}(\mathcal{G}(\cdot))$ to gradually approach the visible mask $M_p$ to complete mask $M_c$. In each step, the input mask is denoised with a stack of generation and segmentation stages. Specifically, as the $\mathcal{S}(\mathcal{G}(\cdot))$ includes a generative process, we can obtain a set of estimations of

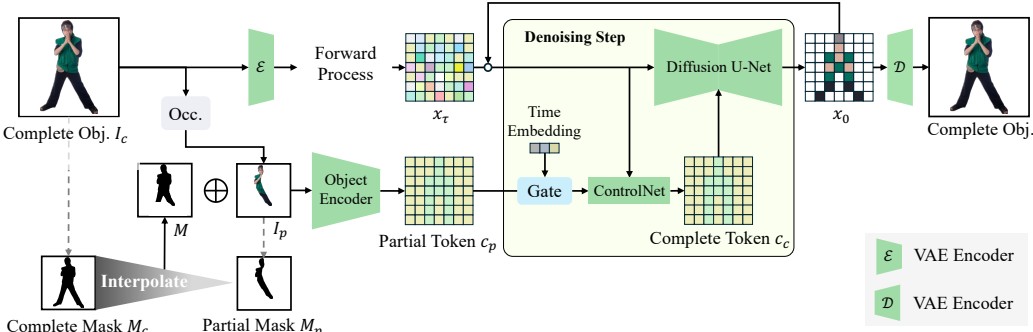

Figure 3: **Illustation of Mask-denoising ControlNet.** The Mask-denoising Controlnet aims to recover the complete object from the partial object and a conditioning mask. Given a complete object $I_c$ and its corresponding mask $M_c$, we first occlude the complete object and keep the visible part as $I_p$. Specifically, we sample a mask $M$ from the interpolations between visible and complete masks as the condition of the generative model during training.

denoised mask $\{M_t^{(i)}\}$. Here, we utilize a function $\mathcal{V}(\cdot)$ to find a more complete and reasonable mask from the $N$ sampled masks and leverage it as the input mask for the next iteration to further denoise. The updating rule can be written as:

$$\hat{M}_t = \mathcal{V}(M_t^{(1)}, \cdots, M_t^{(N)}), \quad \{M_t^{(i)}\}_{i=1}^N = \mathcal{S}(\mathcal{G}(I_p, \hat{M}_{t-1})) \tag{2}$$

where $N$ is the number of sampled images in each iteration. With a satisfactory complete mask $\hat{M}_T$ after $T$ iterations, the object completion can be achieved accordingly by $\mathcal{G}(I_p, \hat{M}_T)$. The mathematical explanation of the process will be discussed in Section 3.3.

## 3.1 GENERATION STAGE

We introduce a mask-denoising ControlNet as the generative model $\mathcal{G}$ for object completion. Different from the conventional object completion methods that solely rely on the visible part of the object, we introduce an additional mask term as the condition.

**Mask as a condition.** In the initial stage of our pipeline, as illustrated on the left side of Fig. 3, we begin with a complete object $I_c$ and its corresponding mask $M_c$. Our approach commences by occluding the complete object, retaining only the partially visible portion as $I_p$. Recall that the mask-denoising procedure initiates with the partial mask $M_p$ and culminates with the complete mask $M_c$. To facilitate this iterative denoising, the model must effectively handle any mask that falls within the interpolation between the initial partial mask and the target complete mask. Consequently, during training, we introduce a mask $M$ obtained from interpolations between the partial and complete masks as a conditioning factor for the generative model.

**Diffusion model.** Diffusion models have achieved notable progress in synthesizing unprecedented image quality and have been successfully applied to many text-based image generation works (Rombach et al., 2022; Zhang et al., 2023). For our object completion task, the complete object can be generated by leveraging the diffusion process.

Specifically, the diffusion model generates image latent $x$ by gradually reversing a Markov forward process. As shown in Figure 3, starting from $x_0 = \mathcal{E}(I_c)$, the forward process yields a sequence of increasing noisy tokens $\{x_\tau | \tau \in [1, T_\mathcal{G}]\}$, where $x_\tau = \sqrt{\bar{\alpha}_\tau} y_0 + \sqrt{1 - \bar{\alpha}_\tau} \epsilon$, $\epsilon$ is the Gaussian noise, and $\alpha_\tau$ decreases with the timestep $\tau$. For the denoising process, the diffusion model progressively denoises a noisy token from the last step given the conditions $c = (I_p, M, E)$ by minimizing the following loss function: $\mathcal{L} = \mathbb{E}_{\tau, x_0, \epsilon} \| \epsilon_\theta(x_\tau, c, \tau) - \epsilon \|_2^2$. $I_p$, $M$, and $E$ are the partial object, conditioned mask, and text prompt respectively.

**Mask-denoising ControlNet.** Previous work (Zhang et al., 2023) has demonstrated an effective way to add additional control to generative diffusion models. We follow this architecture and make

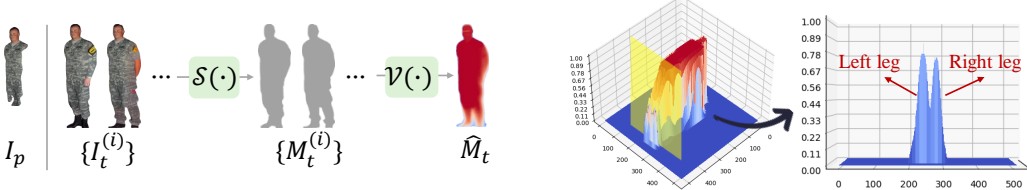

**(a) Illustration of the segmentation stage**      **(b) Visualization of the mask probability map**

Figure 4: We calculate the mask probability map by averaging and normalizing the masks of sampled images. We show a cross-section of the lower leg to better visualize (shown as yellow).

necessary modifications to adapt the architecture to object completion. As shown in Figure 3, given the visible object $I_p$ and the conditioning mask $M$, we first concatenate them and extract the partial token $c_p$ with an object encoder. Different from ControlNet (Zhang et al., 2023) assuming the condition is accurate, the object completion task relies on incomplete conditions. Specifically, in the early diffusion steps, the condition information is vital to complete the object. Nevertheless, in the later steps, inaccurate information in the condition can degrade the generated object. To tackle this problem, we introduce a time-variant gating operation to adjust the importance of conditions in the diffusion steps. We learn a linear transform $f : \mathbb{R}^C \to \mathbb{R}^1$ upon the time embedding $e_\tau \in \mathbb{R}^C$ and then apply it to the partial token as $f(e_\tau) \cdot c_p$ before feeding it to the ControlNet. In this way, the importance of visible features can be adjusted as the diffusion steps forward.

### 3.2 SEGMENTATION STAGE

In the segmentation stage, illustrated in Figure 4 (a), our approach initiates by sampling $N$ images denoted as $\{I_t^{(i)}\}_{i=1}^N$ from the generative model, where $t$ is the IMD step. Subsequently, we employ an off-the-shelf object segmentation model denoted as $\mathcal{S}(\cdot)$ to generate object masks $\{M_t^{(i)}\}$ from these sampled images.

To derive an improved mask for the subsequent IMD step, we seek a function $\mathcal{V}(\cdot)$ that can produce a high-quality mask prediction from the set of $N$ generated masks. In Figure 4 (b), we provide a visualization of the probability map associated with a set of object masks with the same conditions, which is computed by taking the normalized average of the masks. To enhance the visualization of this probability distribution, we focus on a specific cross-section of the fully occluded portion in image $I_p$ (the lower leg, represented as a yellow section) and visualize the probability as a function of the horizontal coordinate which demonstrates an obvious unimodal and symmetric property. Leveraging this observation, we can find an improved mask by taking the high-probability region. The updating can be achieved by conducting a voting process across the $N$ estimated masks, as defined by the following equation:

$$\hat{M}_t[i, j] = \begin{cases} 1, & \text{if } \quad \frac{\sum_{i=1}^N M_t^{(i)}[i,j]}{N} \geq \tau \\ 0, & \text{otherwise} \end{cases} \tag{3}$$

where $[i, j]$ denotes the coordinate, and $\tau$ is the threshold employed for the mask voting process.

### 3.3 DISCUSSION

In this section, we discuss the mathematical explanation of MaskComp, where we will omit the conditioned partial image $I_p$ for simplicity.

**Joint modeling of mask and object.** In practical scenarios where the complete object mask $M_c$ is unavailable, modeling object completion through a marginal probability $p(I_c|M_c)$ becomes infeasible. Instead, it necessitates the more challenging joint modeling of objects and masks, denoted as $p(I, M)$, where the images and masks can range from partial to complete. Let us understand the joint distribution by exploring its marginals. Since the relation between mask and image is one-to-many (each object image only has one mask while the same mask can be segmented from multiple images), the $p(M|I)$ is actually a Dirac delta distribution $\delta$ and only the $p(I|M)$ is a real distribution.

In this way, the joint distribution of mask and image is discrete and complex, making the modeling difficult. To address this issue, we introduce a slack condition to the joint distribution $p(I, M)$ that *the mask and image can follow a many-to-many relation*, which makes its marginal $p(M|I)$ a real distribution and permits $p(I|M)$ to predict an image $I$ that has a different shape as the conditioned $M$ and vice versa.

**Mutual-beneficial sampling.** After discussing the joint distribution that we are targeting, we introduce the mathematical explanation of MaskComp. MaskComp introduces the alternating modeling of two marginal distributions $p(I|M)$ (generation stage) and $p(M|I)$ (segmentation stage), which is actually a Markov Chain Monte Carlo-like (MCMC-like) process and more specifically Gibbs sampling-like. It samples the joint distribution $p(I, M)$ by iterative sampling from the marginal distributions. Two core insights are incorporated in MaskComp: (1) providing a mask as a condition can effectively enhance object generation and (2) fusing the mask of generated object images can result in a more accurate and complete ob-

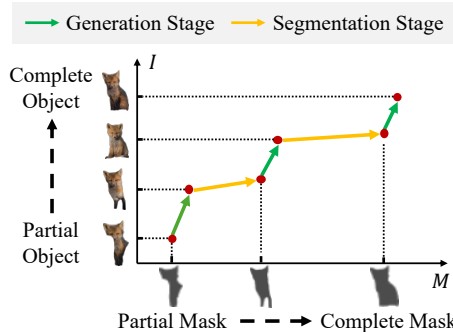

Figure 5: **Mutual-benificial sampling.**

ject mask. Based on these insights, we train Mask-denoising ControlNet to maximize $p(I|M)$ and leverage mask voting to maximize the $p(M|I)$. As shown in Fig. 5, MaskComp develops a mutual-beneficial sampling process from the joint distribution $p(I, M)$, where the object mask is provided to boost the image generation and, in return, the generated images can lead to a more accurate mask by fusing the segmentation of images. Through alternating sampling from the marginal distributions, we can effectively address the object completion task.

## 4 EXPERIMENT

**Dataset.** We evaluate MaskComp on two popular datasets: AHP (Zhou et al., 2021a) and DYCE (Ehsani et al., 2018). AHP is an amodal human perception dataset that is composed of a training set with 56,302 images with annotations of integrated humans, a validation set with 297 images of synthesized occlusion cases, and a test set with 56 images of artificial occlusion cases. As the original test split is too small, we resplit 10,000 images from the training set for evaluation. DYCE is a synthetic dataset with photo-realistic images and the natural configuration of objects in indoor scenes. 41,924 and 27,617 objects are involved in the training set and test sets respectively. For both datasets, the non-occluded ground-truth object and its corresponding mask for each object are available. We train MaskComp on the AHP and a filtered subset of OpenImage v6 (Kuznetsova et al., 2020). OpenImage is a large-scale dataset offering heterogeneous annotations. We select a subset of OpenImage that contains 429,358 objects as a training set of MaskComp.

**Evaluation metrics.** In accordance with previous methods (Zhou et al., 2021a), we evaluate image generation quality Fréchet Inception Distance (FID). As the FID score cannot reflect the object completeness, we further conduct a user study, leveraging human assessment to compare the quality and completeness of images produced by MaskComp and state-of-the-art methods. During the assessment, given a partially occluded object, the participants are required to rank the generated object from different methods based on their completeness and quality. We calculate the averaged ranking and the percentage of the image being ranked as the first place as the metrics.

**Implementation details.** For the generation stage, we train the masked denoising ControlNet with frozen Stable Diffusion (Rombach et al., 2022) on the AHP dataset for 50 epochs. The learning rate is set for 1e-5. We adopt $\mathrm{batchsize} = 8$ and an Adam (Loshchilov & Hutter, 2017) optimizer. The image is resized to $512 \times 512$ for both training and inference. The object is cropped and resized to have the longest side 360 before sticking on the image. We follow (Zhang et al., 2023) to occlude objects. For a more generalized setting, we train the masked denoising ControlNet on a subset of the OpenImage (Kuznetsova et al., 2020) dataset for 36 epochs. We generate text prompts using BLIP (Li et al., 2022) for all experiments (prompts are necessary to train ControlNet). For the segmentation stage, we leverage segment anything model (SAM) (Kirillov et al., 2023) as $\mathcal{S}(\cdot)$. We

| Method | AHP (Zhou et al., 2021a) | | | | DYCE (Ehsani et al., 2018) | | | |
|---|---|---|---|---|---|---|---|---|
| | FID-G ↓ | FID-S ↓ | Rank ↓ | Best ↑ | FID-G ↓ | FID-S ↓ | Rank ↓ | Best ↑ |
| ControlNet | 40.2 | 45.4 | 3.4 | 0.10 | 42.4 | 49.4 | 3.4 | 0.08 |
| Kandinsky 2.1 | 43.9 | 39.2 | 3.2 | 0.11 | 44.3 | 47.7 | 3.4 | 0.06 |
| Stable Diffusion 1.5 | 35.7 | 41.4 | 3.2 | 0.12 | 31.2 | 43.4 | 3.4 | 0.11 |
| Stable Diffusion 2.1 | 30.8 | 39.9 | 3.1 | 0.14 | 30.0 | 41.1 | 3.0 | 0.12 |
| **MaskComp (Ours)** | **16.9** | **21.3** | **2.1** | **0.53** | **20.0** | **25.4** | **1.9** | **0.63** |

Table 1: **Quantitative evaluation on object completion task**. The computing of FID-G and FID-S only considers the object areas within ground truth and foreground regions segmented by SAM, respectively, to eliminate the influence of the generated background. The Rank denotes the average ranking in the user study. The Best denotes the percentage of samples that are ranked as the best. ↓ and ↑ denote the smaller the better and the larger the better respectively.

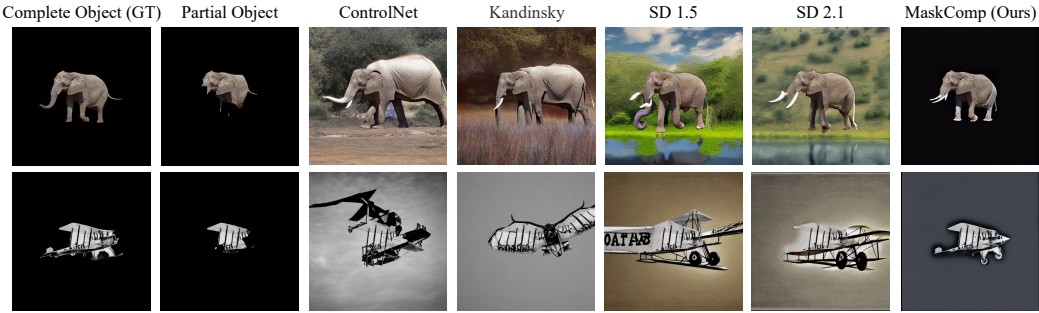

Figure 6: **Qualitative comparison against ControlNet, Kandinsky and Stable Diffusion**. The partial object is the input to the model. The complete object is provided as a good example.

vote mask with a threshold of $\tau = 0.5$. During inference, if no other specification, we conduct the IMD process for 5 steps with $N = 5$ images for each step. We give the class label as the text prompt to facilitate the ControlNet to effectively generate objects. All baseline methods are given the same text prompts during the experiments. During training, we conduct the random occlusion process twice for each complete mask $M_c$. The partial mask $M_p$ is achieved by considering the occluded areas in both of the occlusion processes. The interpolated mask $M$ is generated by using one of the occlusions. The time embedding used for the gating operation is shared with the time embedding for encoding the diffusion step in the stable diffusion. More implementation details are available in the appendix. The code will be made publicly available.

## 4.1 MAIN RESULTS

**Quantitative results.** We compare the MaskComp with state-of-the-art methods (ControlNet (Zhang et al., 2023), Kandinsky 2.1 (Shakhmatov et al., 2023), Stable Diffusion 1.5 (Rombach et al., 2022) and Stable Diffusion 2.1 (Rombach et al., 2022)) on AHP (Zhou et al., 2021a) and DYCE (Ehsani et al., 2018) dataset. The results in Table 1 indicate that MaskComp consistently outperforms other methods, as evidenced by its notably lower FID scores, signifying the superior quality of its generated content. We conducted a user study to evaluate object completeness in which participants ranked images generated by different approaches. MaskComp achieved an impressive average ranking of 2.1 and 1.9 on the AHP and DYCE datasets respectively. Furthermore, MaskComp also generates the highest number of images ranked as the most complete and realistic compared to previous methods. We consider the introduced mask condition and the proposed IMD process benefits the performance of MaskComp, where the additional conditioned mask provides robust shape guidance to the generation process and the proposed IMD process refines the initial conditioned mask to a more complete shape, further enhancing the generated image quality.

**Qualitative results.** We present visual comparisons between MaskComp and Stable Diffusion (Rombach et al., 2022), illustrated in Fig. 6. Our visualizations showcase MaskComp's ability to produce realistic and complete object images given partial images as the condition, whereas previous approaches exhibit noticeable artifacts and struggle to achieve realistic object completion. In

| Mask | Visible | Noisy | Complete |
|---|---|---|---|
| FID | 16.9 | 15.3 | 12.7 |

(a) **Conditioned mask**.

| Occ. | 20% | 40 % | 60 % | 80% |
|---|---|---|---|---|
| FID | 13.4 | 15.7 | 17.2 | 29.9 |

(b) **Occlusion rate**.

| Comp. | Gen. | Segm. | Total |
|---|---|---|---|
| Second | 14.3 | 1.2 | 15.5 |

(c) **Inference time**.

Table 2: **Ablation of MaskComp.** We report the performance with the AHP dataset. (a) We ablate the different conditioning masks during inference. (b) We ablate the occlusion rate during inference. (c) We report the inference time of each component in an IMD step.

| $T$ | 1 | 3 | 5 | 7 |
|---|---|---|---|---|
| FID | 24.7 | 19.4 | 16.9 | 16.1 |

(a) **IMD step number**.

| N | 4 | 5 | 6 |
|---|---|---|---|
| FID | 17.4 | 16.9 | 16.8 |

(b) **# of sampled images**.

| Iter | 20 | 40 | 50 |
|---|---|---|---|
| FID | 16.9 | 15.7 | 15.1 |

(c) **Iter. for diffusion**.

| Gating | ✓ | ✗ |
|---|---|---|
| FID | 16.9 | 18.2 |

(d) **Condition gating**.

Table 3: **Design choices for IMD.** We conduct the experiments on AHP dataset. (a) We ablate the IMD step number. (b) We ablate the number of sampled images in the segmentation stage. (c) We ablate the diffusion iteration for the generative model. (d) We ablate on the gating operation in the mask-denoising ControlNet.

addition, without mask guidance, it is common for previous methods to generate images that fail to align with the partial object.

## 4.2 ANALYSIS

**Performance with different mask conditions.** We conduct ablation studies to investigate the impact of different mask conditions on the generative model's performance. In this analysis, we evaluated the quality of generated images when conditioned on the partial object image along with three distinct types of masks: (1) visible masks, (2) noisy masks, and (3) complete masks characterized by an occlusion level between that of visible and complete masks. As shown in Table 2a, the model achieves its highest performance when it is conditioned with complete object masks, whereas relying solely on visible masks yields less optimal results. These results provide strong evidence that the quality of the conditioned mask significantly influences the quality of the generated images.

**Performance with different occlusion rates.** We perform ablation studies to assess the resilience of MaskComp under varying occlusion levels. As presented in Table 2b, we evaluate MaskComp across object occlusion rates ranging from 20% to 80%, where the occlusion rate represents the proportion of the obscured area compared to the complete object. The results indicate that MaskComp's performance declines only slightly as occlusion rates rise. Even at 60% occlusion rates, its robust performance holds up. However, a further increase in the occlusion rate to an extreme level will result in MaskComp not producing high-quality images.

**Inference time.** We demonstrate the inference time of each component in IMD as shown in Table 2c (with a single NVIDIA V100 GPU). Due to the multiple diffusion processes in each IMD step, the inference speed of MaskComp is slow. To improve the inference speed, we notice that decreasing the diffusion step number in the first several IMD steps will not severely degrade the performance. By incorporating this idea into MaskComp, the average running time was reduced to 2/3 original time with a slight FID increase of 0.5.

**Design choices in IMD.** We conduct experiments to ablate the design choices in IMD and their impacts on the completion performance. We first study the effect of IMD step number. With a larger step number, IMD can better advance the partial mask to the complete mask. As shown in Table 3a, we notice that the image quality keeps increasing and slows down at a step number of 5. In this way, we choose 5 as our IMD step number. After that, we ablate the number of sampled image in the segmentation stage in Table 3b. We notice more sampled images generally leading to a better performance. We leverage an image number of 5 with the efficiency consideration. We ablate the iterations for the diffusion process. Table 3c demonstrates that a larger diffusion iteration number can lead to a better performance which is as expected. In addition, as the input condition for the object completion task is not accurate, we introduce a time-variant gating operation to facilitate the

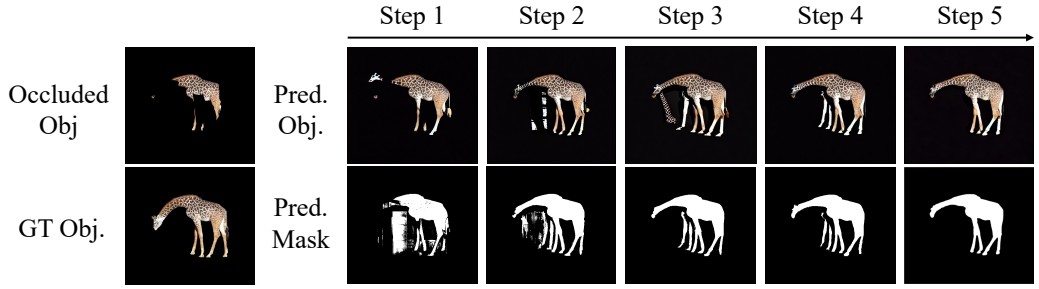

Figure 7: **Visualization of the IMD process.** For each step, we randomly demonstrate one generated image and the averaged mask for all generated images. We omit the input mask which has the same shape as the input occluded object.

generation process. As shown in Table 3d, we notice the gating operation improves the generation quality by 1.3 FID, indicating the necessity of conditional gating.

**Visualization of iterative mask denoising.** To provide a clearer depiction of the iterative IMD process, as depicted in Fig. 7, we present visualizations of the generated image and the averaged mask for each step. In the initial step, we observe the emergence of artifacts alongside the object. As we progress through the steps, both the image and mask quality exhibit continuous improvement.

**Failure case analysis.** Despite the robust capabilities of the Mask-denoising ControlNet and SAM models, they can still generate low-quality images and inaccurate segmentation results. In Fig. 13, we show a case where the intermediate stage of IMD produces a human with an extra right arm. To address this, we implement three key strategies: (1) **Error Mitigation during Segmentation with SAM**: As shown in Fig. 13, SAM effectively filters out incorrectly predicted components, such as a misidentified right arm, resulting in a more coherent shape for subsequent iterations. SAM's robust instance understanding capability extends to not only accurately segmenting objects with regular shapes but also filtering out irrelevant parts when additional objects/parts are generated. (2) **Error Suppression through Mask Voting**: In cases where only a few generated images exhibit errors, the impact of these errors can be mitigated through mask voting. The generated images are converted to masks, and if only a minority display errors, their influence is diminished through the voting operation. (3) **Error Tolerance in IMD Iteration**: We train the mask-denoising ControlNet to handle a wide range of occluded masks. Consequently, if the conditioned mask undergoes minimal improvement or degradation due to the noises in a given iteration, it can still be improved in the subsequent iteration. While this may slightly extend the convergence time, it is not anticipated to have a significant impact on the ultimate image quality. More analysis is available in the Appendix.

More ablation studies and analyses are available in the Appendix.

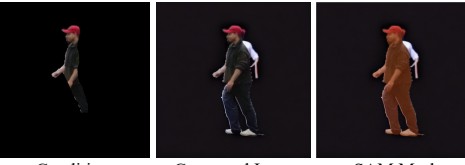

Figure 8: **Failure case.**

# 5 CONCLUSION

In this paper, we introduce MaskComp, a novel approach for object completion. MaskComp addresses the object completion task by seamlessly integrating conditional generation and segmentation, capitalizing on the crucial observation that the quality of generated objects is intricately tied to the quality of the conditioned masks. We augment the object completion process with an additional mask condition and propose an iterative mask denoising (IMD) process. This iterative approach gradually refines the partial object mask, ultimately leading to the generation of satisfactory objects by leveraging the complete mask as a guiding condition. Our extensive experiments demonstrate the robustness and effectiveness of MaskComp, particularly in challenging scenarios involving heavily occluded objects.

| Model | Mask2Former | ClipSeg | SAM |
|-------|-------------|---------|-----|
| FID | 22.5 | 19.9 | 16.9 |

(a) **Segmentation model** $\mathcal{S}$.

| Strategy. | Logits (V) | Logits (M) | Mask (V) | Mask (M) |
|-----------|-----------|-----------|----------|----------|
| FID | 16.9 | 17.2 | 17.6 | 17.0 |

(b) **Voting strategies**.

| Method | AISFormer+ControlNet | MaskComp |
|--------|----------------------|----------|
| FID | 29.4 | 16.9 |

(c) **Amodal baseline**.

| Occ. | Rectangle | Oval | Object |
|------|-----------|------|--------|
| FID | 15.3 | 15.1 | 16.9 |

(d) **Occlusion type**.

Table 4: **More ablation of MaskComp.** We report the performance with the AHP dataset. (a) We ablate the segmentation model. (b) We ablate voting strategies. V: voting. M: Mean. (c) We report the performance compared to the amodal segmentation baseline. (d) We report the performance with different types of occlusion.

## A    MORE EXPERIMENTS

In this section, we provide more ablation experiments and analysis of MaskComp. We conducted ablation experiments to determine the design choice in the segmentation stage. We report the ablation studies about segmentation models and voting strategies in Table 4a and Table 4b. We notice SAM and voting with logits achieve the best performance. The current design choice of using SAM and voting with logits is based on the ablation results. In addition, a reasonable baseline to compare is generating objects using ControlNet with an amodal segmentation model to generate a conditioned mask. We leverage the state-of-the-art amodal segmentation AISFormer Tran et al. (2022) to provide masks and generate corresponding objects using ControlNet as shown in Table 4c. We notice that MaskComp achieves an obviously better performance compared to the baseline. To understand the influence of occlusion type, we conduct an ablation study as shown in Table 4d. We notice that the occlusion with a more complicated object shape will impose more challenges on the proposed model.

## B    MORE DISCUSSION

| Type | Noise | Network | Target |
|------|-------|---------|--------|
| Image diffusion | Gaussion | UNet | Predict added noise |
| Mask denoising | Occlusion | Mask denoiser $\mathcal{S}(\mathcal{G}(\cdot))$ | Predict denoised mask |

Table 5: **Analogy between image diffusion and mask denoising**.

**Image diffusion *v.s.* Mask denoising.**    During the training of the image diffusion model, Gaussian noise is introduced to the original image. A denoising U-Net is then trained to predict this noise and subsequently recover the image to its clean state during inference.

Similarly, in the context of the proposed iterative mask denoising (IMD) process, we manually occlude the complete object (which can be assumed as adding noise) and train a generative model to recover the complete object. During inference, as shown in Eq. (1), we employ an iterative approach that combines the segmentation and generation model $\mathcal{S}(\mathcal{G}(\cdot))$ functioning as a denoiser. This denoiser progressively denoises the partial mask to achieve a complete mask, following a similar principle to the denoising diffusion process. By drawing parallels between image diffusion and mask denoising, we establish an analogy, as depicted in Table 5. We can notice that the mask denoising process shares the spirits of the image diffusion process and the only difference is that mask denoising does not explicitly calculate the added noise but directly predicts the denoised mask. In this way, MaskComp can be assumed as a double-loop denoising process with an inner loop for image denoising and an outer loop for mask denoising.

**Training without complete object.**    In the context of image diffusion, though multiple forward steps are involved to add noise to the image, the network only learns to predict the noise added in a single step during training. Therefore, if we possess a set of noisy images generated through

|  | Input | Step 1 | Step 2 | Step 3 | Step 4 | Step 5 |

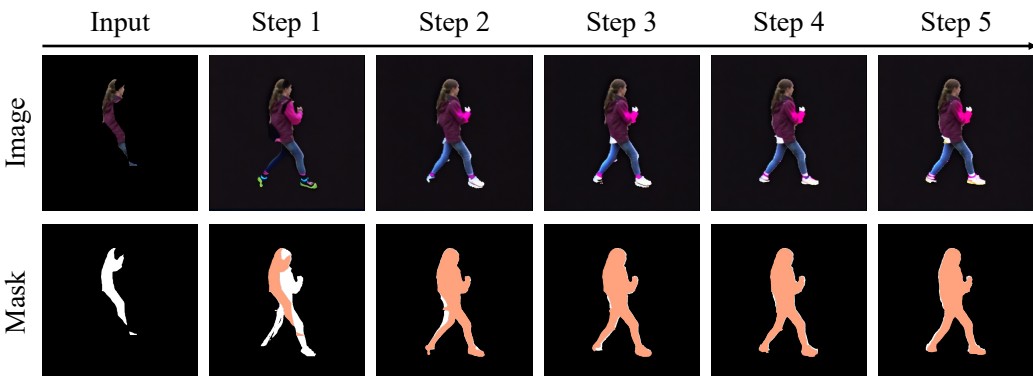

Figure 9: **Visualization of IMD process with model trained without complete objects**. To better visualize the iterative mask denoising process, we denote the overlapping masked area from the last iteration as orange. We can notice that the object shape is gradually refined and converged to a complete shape.

forward steps, the original image is not required during the training. This motivates us to explore the feasibility of training MaskComp without relying on the complete mask. Similar to image diffusion, given a partial mask, we can further occlude it and learn to predict the partial mask before further occlusion. In this way, MaskComp can be leveraged in a more generic scenario without the strict demand for complete objects. We have discussed the quantitative results in Section 4.2. Here, we visualize the IMD process with a model trained without complete objects (on OpenImage). To better visualize the object shape updating, we denote the overlapping masked area from the last step as orange. We can notice that the object shape gradually refines and converges to the complete shape as the IMD process forwards. Interestingly, the IMD process can learn to complete the object even if only a small portion of the complete object was available in the dataset during the training. We consider this property to make it possible to further generalize MaskComp to the scenarios in which a complete object is not available.

**What will the marginal distribution $p(I|M)$ and $p(M|I)$ be like without the slack condition?** The relation between mask and object image is one-to-many. The $p(I|M)$ models a filling color operation that paints the color within the given mask area. And as each object image only corresponds to one mask, the $p(M|I)$ is a deterministic process that can be modeled by a delta function $\delta$. Previous methods generally leverage the unslacked setting. For example, the ControlNet assumes the given mask condition can accurately reflect the object shape and therefore, it can learn to fill colors to the masked regions.

**Background objects in the generated images.** The training of mask-denoising ControlNet aims to learn an intra-object correlation. We leverage a black background to eliminate the influence of background objects. However, we notice that even if we train the network with the black background as ground truth, it is still possible to generate irrelevant objects in the background. As shown in Fig. 10, we visualize an image that generates a leather bag near the women. We consider the generated background object can result from the learned inter-object correlation from the frozen Stable Diffusion model Rombach et al. (2022). As the generated background object typically will not be segmented in the segmentation stage, it will not influence the performance of MaskComp.

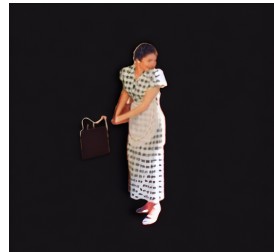

Figure 10: **BG objects.**

**Potential applications.** Object completion is a fundamental technique that can boost a number of applications. For example, a straightforward application is the image editing. With the object completion, we can modify the layer of the objects in an image as we modify the components in the PowerPoint. It is possible to bring forward and edit objects as shown in Fig. 11. In addition, object

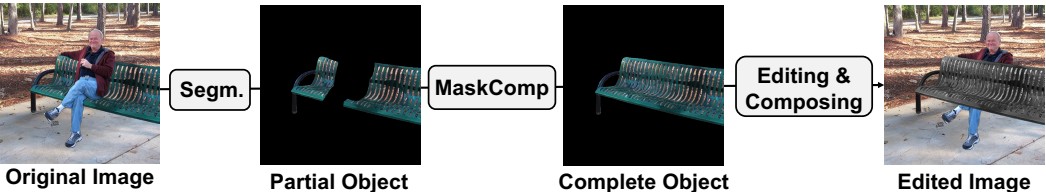

Figure 11: **Illustation of potential application**.

completion is also an important technique for data augmentation. We hope MaskComp can shed light on more applications leveraging object completion.

## C  MORE EXPERIMENTS

**More implementation details.**    We leverage two types of occlusion strategies during the training of mask-denoising ControlNet. First, we randomly sample a point on the object region, and then randomly occlude a rectangle area with the sampled point as the centroid. The width and height of the rectangle are determined by the width and height of the bounding box of the ground truth object. We uniformly sample a ratio within [0.2, 0.9] and apply it to the ground truth width and height to occlude the object. Second, we randomly occlude the object by shifting its mask. Specifically, we randomly shift its mask by a range of [0.17, 0.25] and occluded the region within the shifted mask. We equally leverage these two occlusion strategies during training. For the object encoder to extract partial token $c_p$ in the mask-denoising ControlNet, we utilize a Swin-Transformer Liu et al. (2021) pre-trained on ImageNet Deng et al. (2009) with an additional convolution layer to accept the concatenation of mask and image as input. We initialize the mask-denoising ControlNet with the pre-trained weight of ControlNet with additional mask conditions. To segment objects in the segmentation stage, we give a mix of box and point prompts to the Segment Anything Model (SAM). Specifically, we uniformly sample three points from the partial object as the point prompts and we leverage an extended bounding box of the partial object as the box prompts. We also add negative point prompts at the corners of the box to further improve the segmentation quality.

**More visualization.**    As shown in Fig. 12, we provide more qualitative comparisons with Stable Diffusion (Rombach et al., 2022). We notice that Stable Diffusion tends to complete irrelevant objects to the complete parts and thus leads to an unrealism of objects. Instead, MaskComp is guided by a mask shape and successfully captures the correct object shape thus achieving superior results.

**Failure case analysis.**    We present a failure case in Fig. 13, where MaskComp exhibits a misunderstanding of the pose of a person bending over, resulting in the generation of a hat at the waist. We attribute this generation of an unrealistic image to the uncommon pose of the partial human. Given that the majority of individuals in the AHP training set have their heads up and feet down, MaskComp may have a tendency to generate images in this typical position.

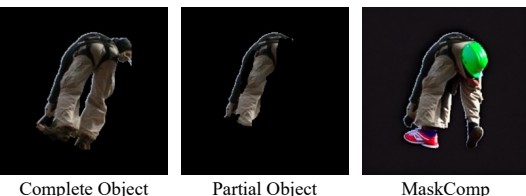

Complete Object     Partial Object     MaskComp

Figure 13: **Failure case.**

We consider that with a more diverse dataset, including images of individuals in unusual poses, MaskComp could potentially yield superior results in handling similar cases.

**Details of user study.**    There are 16 participants in the user study. All participants have relevant knowledge to understand the task. During the assessment, each participant is provided with instructions and an example to understand the task. We show an example of the images presented during the user study as Fig. 14 and Fig. 15. We list the instructions as follows.

Task: Given the partial object (lower left), generate the complete object (upper left).

455   Instruction:

456        • Ranking images 1-5, put the best on the left and the worst on the right.

457        • Please focus on the foreground object and ignore the difference presented in the back-
458          ground.

459        • Original image is provided as a good example.

460        • The criteria for ranking are founded on object quality, encompassing aspects such as com-
461          pleteness, realism, sharpness, and more.

462        • It must be strictly ordered (no tie).

463        • Please rank the image in the following form: 1;2;3;4;5 or 5;4;3;2;1 (Use a colon to separate,
464          no space at the beginning)

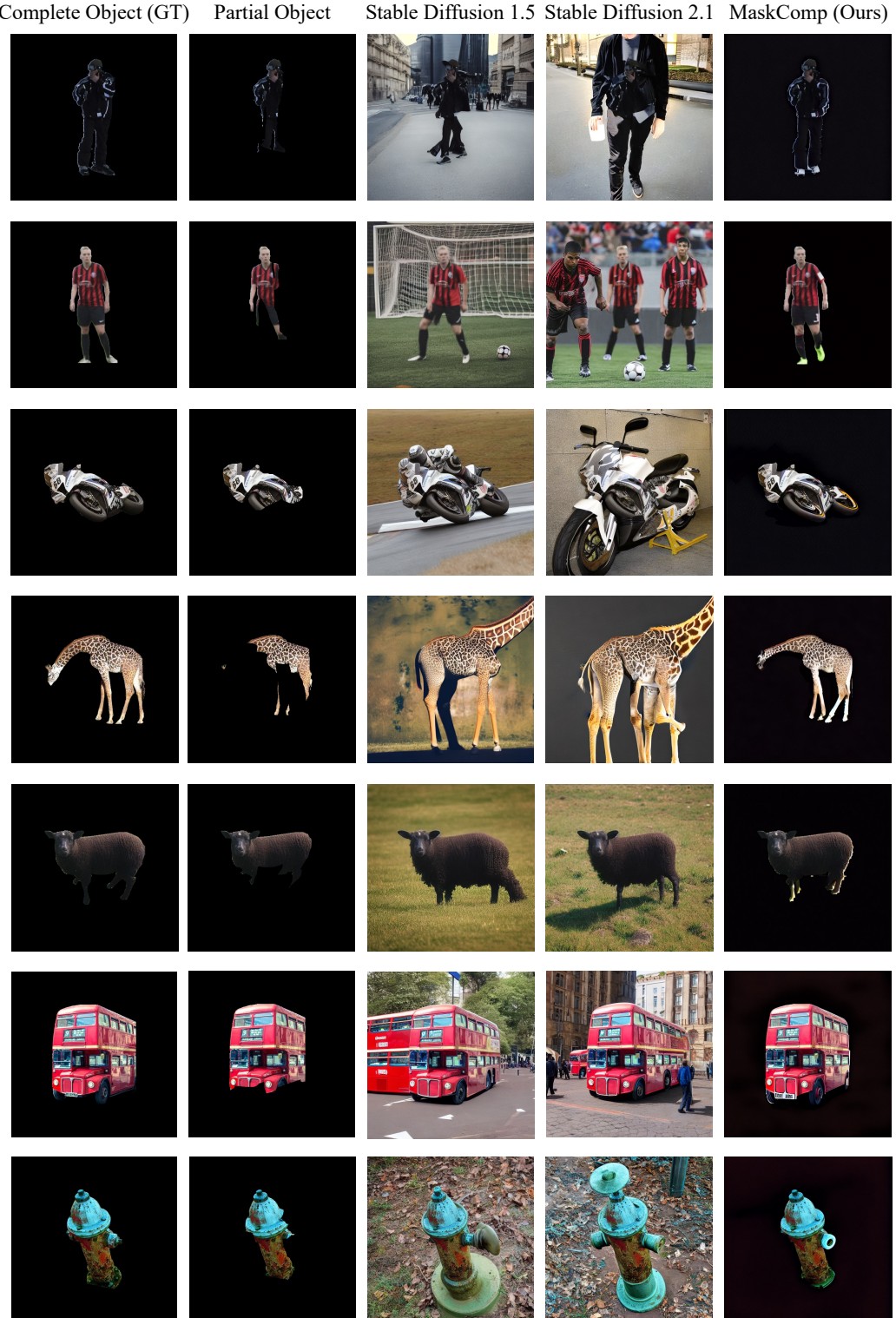

Figure 12: **More qualitative comparison with Stable Diffusion (Rombach et al., 2022).**

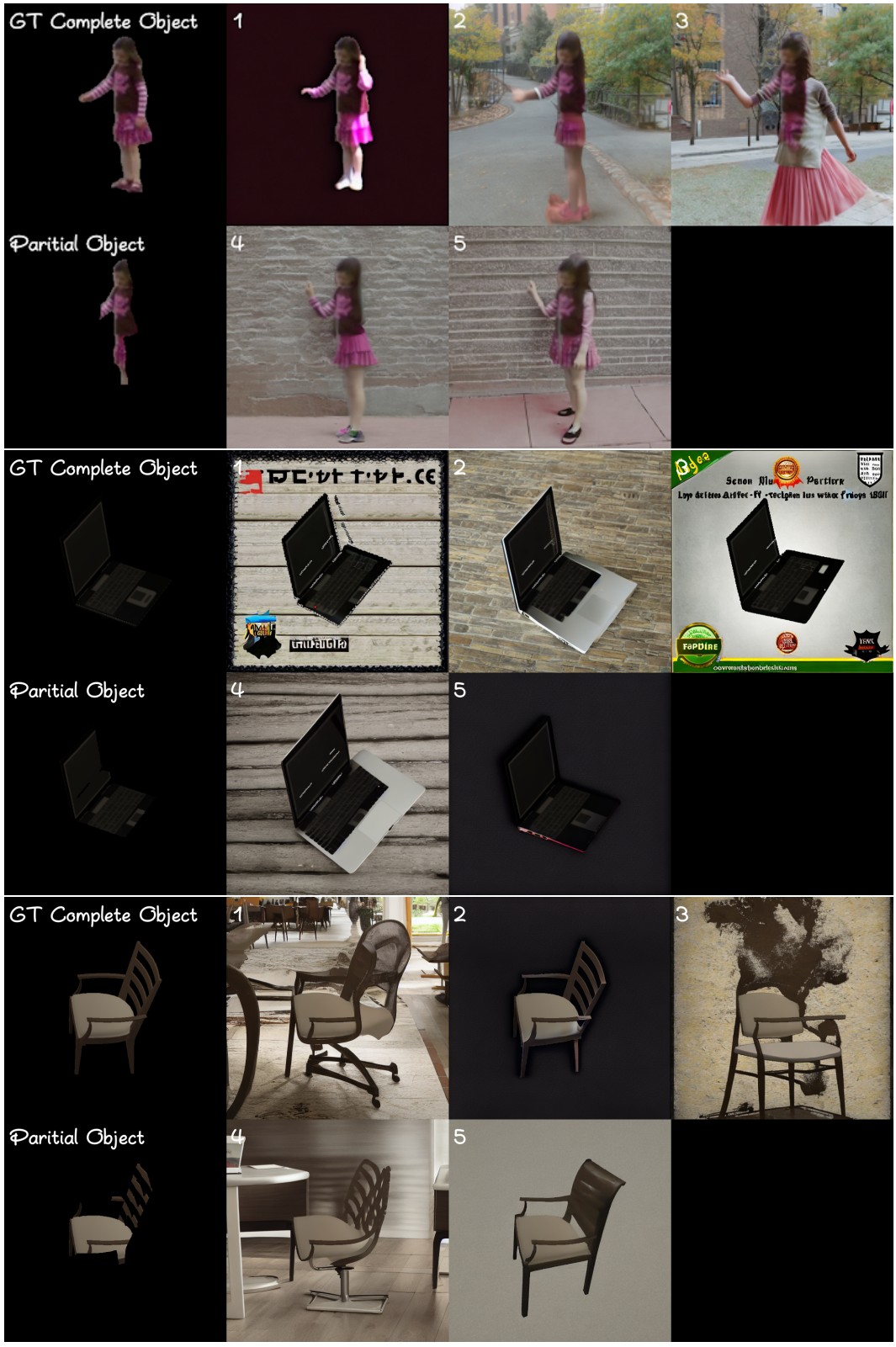

Figure 14: **Examples presented during the user study**.

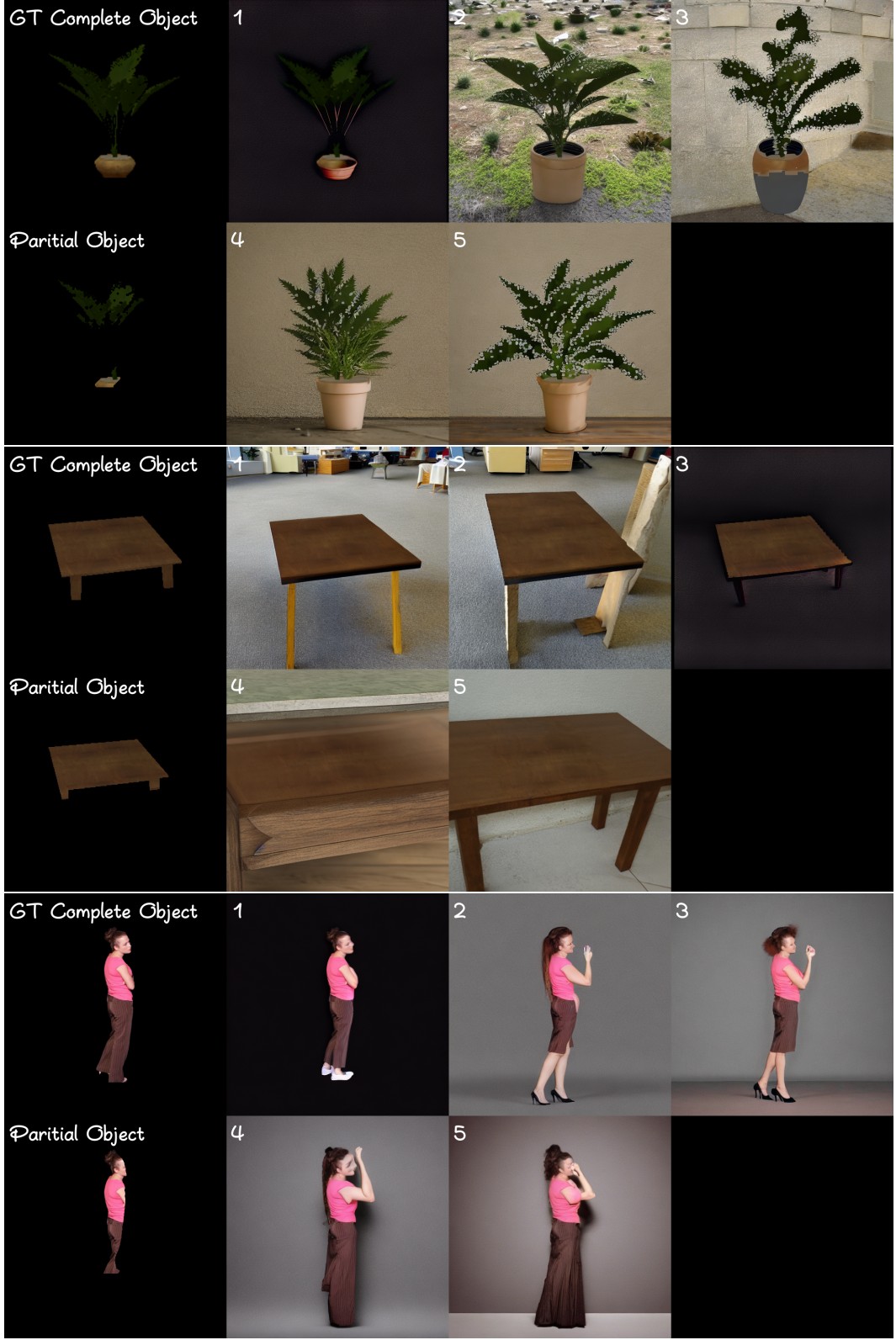

Figure 15: **Examples presented during the user study**.

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
