# OpenReview forum: "Completing Visual Objects via Bridging Generation and Segmentation"
_ICLR.cc/2024/Conference — Submitted to ICLR 2024_

### Official Review · Reviewer_t7dr · 2023-10-28

**Soundness:** 2 fair
**Presentation:** 2 fair
**Contribution:** 3 good
**Rating:** 6
**Confidence:** 5

**Summary:**

This paper proposes to boost the object completion via integrating object segmentation into the denoising process of diffusion. The listed visual results look good.

**Strengths:**

Introducing mask segmentation to facilitate object completion is reasonable, since the completed objects shouldn't reflect strange shapes.

**Weaknesses:**

1. Overall, the paper is easy to follow, but need to be further improved. For example, some symbols are not well explained like the condition $E$ in Line 156. The subscript '_t' is misleading to denote the denoising step of DDPM/DDIM and the proposed IMD. The figure 3 is also confusing. The diffusion model should denoise $x_t$ to $x_0$, but the authors give the start noise as $x_0$. Besides, the time-step of IMD is not illustrated in this figure.

2. It's not rigorous to name the progressive completion process as 'mask denoising'. The object's mask generated from SAM is taken as  a condition to diffusion instead of the denoising uint as $x_t$.


3. It is not clear that whether the other comparison method are retrained in the evaluation data. If yes, the training details and the incomplete mask's interaction in their networks should be claimed, otherwise the authors should explain how to obtain these completion results with the off-the-shelf generation models.

4. It would be better if the authors can give some analysis on the sampling speed and more qualitative results of the proposed method.

**Questions:**

Overall, the proposed idea of leveraging object segmentation to boost object completion is interesting, while the authors should further improve the manuscript to make the contribution more convincing. I am glad to upgrade the rate depending on the rebuttal.

---

> ### Author Response · Authors · 2023-11-15
> **Response to Reviewer t7dr**
>
> We thank the reviewer for the time and effort to review our paper. Our answers to the questions are as follows.
>
> ---
>
> **1. Some symbols in the paper need to be further improved.**
>
> We would like to thank the reviewer for the detailed inspection. We have incorporated the suggested changes to the revised manuscript.
>
> ---
>
> **2. It's not rigorous to name the progressive completion process as 'mask denoising'. The object's mask generated from SAM is taken as a condition to diffusion instead of the denoising unit as $x_t$.**
>
> Thanks for the comments. We agree that the mask denoising is very different from the image denoising diffusion while we hope the word "mask denoising" can help the reader understand our iterative mask completion process. Given a partially visible object $I_p$ and its corresponding partial mask $M_p$, the conventional object completion task aims to find a generative model $\mathcal{G}$ such that $I_c\leftarrow\mathcal{G}(I_p)$, where $I_c$ is the complete object. Here, we additionally add the partial mask $M_p$ to the condition  $I_c\leftarrow\mathcal{G}(I_p, M_p)$, where $M_p$ can be assumed as an addition of the complete mask and a noise $M_p=M_c+\Delta$. By introducing a segmentation model $\mathcal{S}$, we can find a mask denoiser $\mathcal{S}\circ\mathcal{G}$ from the mask completion process: $M_c\leftarrow\mathcal{S}\circ\mathcal{G}(I_p, M_c+\Delta)$. If we consider the occlusion $\Delta$ as a noise, $\mathcal{S}\circ\mathcal{G}$ can be assumed as conducting mask denoising for each IMD step. We will further clarify the usage of "mask denoising" in the revision.
>
> ---
>
> **3. It is not clear whether the other comparison methods are retrained in the evaluation data. If yes, the training details and the incomplete mask's interaction in their networks should be claimed, otherwise the authors should explain how to obtain these completion results with the off-the-shelf generation models.**
>
> We would like to clarify that the baseline methods are not retrained in the evaluation dataset. We consider the baselines can be directly utilized due to (1) the object completion can be treated as a subtask of the in/outpainting objectives that the baseline methods are trained on (2) the object categories in the evaluation set are common which should be within the large-scale training data of the baselines. In addition, to evaluate on a more general object setting, our model is also not trained on the evaluation set (DYCE) but on OpenImage (Line 231). In this way, we consider our comparison can fairly reflect the effectiveness of the proposed method.
>
> ---
>
> **4. It would be better if the authors can give some analysis on the sampling speed and more qualitative results of the proposed method.**
>
> |Reduce SD step|Generation stage|Segmentation Stage|Total|FID|
> | :----:| :----: | :----: | :----: | :----: |
> || 14.3s | 1.2s |15.5s |16.9|
> |$\checkmark$| 8.6s | 1.2s|9.8s| 17.4|
>
> Table D: Inference time for each stage of IMD on a single V100 GPU.
>
> Thanks for your suggestion. We provide the sampling time of each component as shown in Table D. We notice that the most time-consuming part is the diffusion process. To improve the inference speed, we notice that decreasing the diffusion step number in the first several IMD steps will not severely degrade the performance. By incorporating this idea into MaskComp, the average running time was reduced to 2/3 original time. We also provide more qualitative results and results in a potential application (layered image generation) as shown in Figure A ([github.com/iclr23anonymous739/more_vis.pdf](https://github.com/iclr23anonymous739/rebuttal/blob/main/more_vis.pdf)) and Figure B ([github.com/iclr23anonymous739/application.pdf](https://github.com/iclr23anonymous739/rebuttal/blob/main/application.pdf)). We believe the idea of using segmentation to boost generation and the strong capability of MaskComp can be of interest to the community. We will incorporate the suggested results in the revision.

---

> > ### Comment · Reviewer_t7dr · 2023-11-20
> > **Response to authors**
> >
> > I am glad to see the authors' feedback.
> > After reading the comments from other reviewers, some new concerns arose.
> >
> > 1. The current analysis of limitations is insufficient. Is it possible that the $N$ generated candidates are with error generations? The object masks by SAM cannot tell the semantic correctness of the candidates. The authors should give a more sufficient analysis of the failure cases.
> >
> > 2. The reasons for comparing the finetuned diffusion with the other off-the-shelf competitors for ''fair comparison'' are not convincing. If the finetuning process is not necessary for other methods due to their original powerful generalization ability, then what is the reason for the proposed method to conduct the retraining process? Although not trained on DYCE,  the used training datasets including AHP and OpenImage usually have diverse samples. The authors should demonstrate that the superiority of the proposed method is from the help of segmentation instead of the retraining process.

---

> > > ### Author Response · Authors · 2023-11-20
> > > **Response to Reviewer t7dr**
> > >
> > > We thank the reviewer for the additional comments. Our answers to the questions are as follows.
> > >
> > > ---
> > >
> > > **1. Is it possible that the generated candidates are with error generations? The object masks by SAM cannot tell the semantic correctness of the candidates.**
> > >
> > > Yes, it is possible that the generated candidates contain errors. To illustrate, we present a specific failure case observed in the intermediate step of the IMD process, detailed in Figure C ([github.com/iclr23anonymous739/failure_analysis.pdf](https://github.com/iclr23anonymous739/rebuttal/blob/main/failure_analysis.pdf)). We identify two key mechanisms designed to mitigate errors in the generation process:
> > > - *Error Suppression through Mask Voting*: In cases where only a few generated images exhibit errors, the impact of these errors can be mitigated through mask voting. The generated images are converted to masks, and if only a minority display errors, their influence is diminished through the voting operation.
> > > - *Error Mitigation during Segmentation with SAM*: SAM plays a crucial role in the segmentation process. It aids in mitigating certain types of errors, as illustrated in our visualized case. SAM effectively filters out incorrectly predicted components, such as a misidentified right arm, resulting in a more coherent shape for subsequent iterations. SAM's robust instance understanding capability extends to not only accurately segmenting objects with regular shapes but also filtering out irrelevant parts when additional objects/parts are generated.
> > >
> > > While these mechanisms enhance the error-handling capacity of MaskComp in the majority of cases, we acknowledge the existence of scenarios leading to failure. As demonstrated in Figure 12 ([github.com/iclr23anonymous739/failure.pdf](https://github.com/iclr23anonymous739/rebuttal/blob/main/failure.pdf)), there are instances where MaskComp struggles to generate a realistic object, failing to comprehend the correct pose in this particular case. Our ongoing efforts include refining the model to address such challenges and further improve its overall performance.
> > >
> > > ---
> > >
> > > **2. What is the reason for the proposed method to conduct the retraining process? The authors should demonstrate that the superiority of the proposed method is from the help of segmentation instead of the retraining process.**
> > >
> > > |T|1|3|5|7|
> > > | ----| :----: | :----: | :----: | :----: |
> > > |FID| 24.7| 19.4| 16.9 |16.1|
> > >
> > > Table 3 (a): Performance with different IMD steps.
> > >
> > > |Method|ControlNet|Kandinsky 2.1|SD 1.5|SD 2.1|MaskComp w./o. Segm.|
> > > | ----| :----: | :----: | :----: | :----: |:----: |
> > > |FID| 40.2 | 39.2| 35.7 |30.8|33.4|
> > >
> > > Table E: Performance comparison without the facilitation of segmentation.
> > >
> > > Thanks for the feedback. Unlike previous mask-conditioned generation methods (e.g. ControlNet) using complete conditioned masks,  MaskComp employs partial conditioned masks for generating complete objects and proposes the IMD process to facilitate the generation process. We conduct retraining and adjustments to adapt the model to partial masks.
> > >
> > > In Table 3 (a), we present ablation results demonstrating the performance improvement with increasing IMD steps (interleaved generation and segmentation). Additionally, we report performance at step 0 (without segmentation facilitation) and compare it with baseline results in Table E. Notably, without segmentation facilitation, MaskComp exhibits minimal improvement. However, with segmentation facilitation, a significant enhancement over baselines is observed, emphasizing the crucial role of segmentation in our method's effectiveness.
> > >
> > > ---
> > >
> > > We hope the responses could clarify your concerns, and we are more than happy to provide further explanations. Thank you.

---

> > > > ### Comment · Reviewer_t7dr · 2023-11-21
> > > > **Response to authors**
> > > >
> > > > Thanks for your response.
> > > > I advise the authors to give a sufficient analysis on the failure cases in the next version.
> > > > As for the comparison with other diffusion models, two questions still remain.
> > > >
> > > > 1. How to process the occluded mask as a condition to generate the completed result?
> > > >
> > > > 2. Whether the proposed MaskComp is only effective at the retrained controlnet or can be directly used in the pretrained version.

---

> > > > > ### Author Response · Authors · 2023-11-21
> > > > > **Response to Reviewer t7dr**
> > > > >
> > > > > We thank the reviewer for the valuable suggestion. We have incorporated the discussed failure case analysis into the updated version (Line 319-338). Our answers to the questions are as follows.
> > > > >
> > > > > ---
> > > > >
> > > > > **1. How to process the occluded mask as a condition to generate the completed result?**
> > > > >
> > > > > We have three main steps to process the conditioned mask and partial image before feeding them to the diffusion model to generate complete results.
> > > > >
> > > > > - *Occlusion*: As shown in Figure 3 ([github.com/iclr23anonymous739/pipeline.pdf](https://github.com/iclr23anonymous739/rebuttal/blob/main/pipeline.pdf)), given a complete object and its mask, we first create the partial image $I_p$ and a conditioned mask $M$ by adding occlusion  (the details of occlusion strategies can be found in Line 527-534).
> > > > >
> > > > > - *Feature Extraction*: We leverage a vision transformer as the object encoder to encode the conditioned object information. Specifically, taking the concatenation of the partial image $I_p$ and the interpolated mask $M$ as input, the object encoder outputs a partial token $c_p$ (the shape of $I_p$ and $M$ can be different).
> > > > >
> > > > > - *Time-variant Gating*: Unlike ControlNet assuming the condition is accurate, the object completion task relies on incomplete conditions. Specifically, in the early diffusion steps, the condition information is vital to complete the object. Nevertheless, in the later steps, inaccurate information in the condition can degrade the generated object. To tackle this problem, we introduce a time-variant gating operation to adjust the importance of conditions in the diffusion steps. We learn a linear transform $f: \mathbb{R}^{C}\rightarrow\mathbb{R}^{1}$ upon the time embedding $e_\tau\in\mathbb{R}^{C}$ and then apply it to the partial token as $f(e_\tau)\cdot c_p$ before feeding it to the ControlNet. In this way, the importance of partial features can be adjusted as the diffusion steps forward. (We verified the effectiveness of the gating in Table 3 (d).)
> > > > >
> > > > > We use the complete object as the ground truth to train the diffusion model. This supervises the model to generate a complete object from a partial one, alongside a conditioned mask. In addition, the model also operates iteratively through the designed IMD process, enhancing image quality with each iteration.
> > > > >
> > > > > ---
> > > > >
> > > > > **2. Whether the proposed MaskComp is only effective at the retrained controlnet or can be directly used in the pretrained version.**
> > > > >
> > > > > |Method|ControlNet|MaskComp|
> > > > > | ----| ----| ---- |
> > > > > |Target| $I\leftarrow \mathcal{G}(I_p, M)$ |$I_c\leftarrow \mathcal{G}(I_p, M_p)$|
> > > > > |Target with Segm.| $M\leftarrow \mathcal{S}\circ\mathcal{G}(I_p, M)$ |$M_c\leftarrow \mathcal{S}\circ\mathcal{G}(I_p, M_p)$|
> > > > >
> > > > > Table F: Objectives comparison. $\mathcal{G}$: generative model. $\mathcal{S}$: SAM. Subscript $_c$: complete. Subscript $_p$: partial.
> > > > >
> > > > > We would like to clarify that the retraining is necessary. As shown in Table F, we compare the training targets of ControlNet and MaskComp. ControlNet assumes the conditioned mask is accurate and aims to generate image $I$ that aligns with the given mask shape $M$. In this way, even with the facilitation of segmentation model $\mathcal{S}$, the mask cannot be improved with ControlNet $M\leftarrow \mathcal{S}\circ\mathcal{G}(I_p, M)$.
> > > > >
> > > > > However, as analyzed in Line 118-129, the IMD process requires a model that can iteratively complete the partial mask $M_p$ to the complete mask $M_c$. As discussed in question 1, the training objective of our model is to generate a complete image based on partial mask: $I_c\leftarrow \mathcal{G}(I_p, M_p)$. With the facilitation of a segmentation model, the improved mask can be achieved $M_c\leftarrow \mathcal{S}\circ\mathcal{G}(I_p, M_p)$.
> > > > >
> > > > > ---
> > > > >
> > > > > We appreciate the time the reviewer has dedicated to the discussion and hope our response can address the concerns. Thank you.

---

> ### Comment · Reviewer_t7dr · 2023-11-21
> **Response to authors**
>
> Thanks again for the authors feedback. My concerns have been resolved. I am glad to update my rate.

---

### Official Review · Reviewer_2Pq9 · 2023-10-31

**Soundness:** 2 fair
**Presentation:** 3 good
**Contribution:** 2 fair
**Rating:** 5
**Confidence:** 4

**Summary:**

The paper presents an iterative generation strategy for object completion, which alternates between a mask-conditioned image generation stage and a segmentation stage. Specifically, given a partial object and partial mask, the generation stage trains a conditional diffusion U-Net to generate a complete object; the segmentation stage re-segments the generated object image, and the result, which hopefully is more complete,  will be used as the conditional mask for the subsequent generation stage.  The proposed method is evaluated on AHP and DYCE datasets with comparisons to diffusion-based baselines.

**Strengths:**

- The proposed joint object image and mask completion strategy is well-motivated and the overall method seems novel for the target task.
- The paper is mostly easy to follow.
- The experiments include both automatic and human-based metrics for evaluation, and the results are better than baselines.

**Weaknesses:**

- The justification for the entire iterative procedure is lacking. While it is ideal to achieve improvements as shown in Figure 5, there is no guarantee that such improvement can be realized in a realistic setting. In particular, the mask-denoising controlnet is trained in a local manner, which may generate a worse image, and the segmentation stage is largely dependent on the segmentation model S, which may produce noisy segmentation output. Therefore, it is unclear whether this design would work in general cases.

- The proposed method lacks mode diversity for object completion. The segmentation stage uses an averaging operator, which seems problematic since it would lead to mode average and is unable to capture potential different modes in object completion. This is important to generate different candidates for image editing since there are multiple possibilities for the occluded regions.

- Some technical aspects of the method are unclear. For example:
   + In the generation stage, how does the interpolation is implemented? How does the method generate the interpolated masks and its time embedding?
   + What are the notations M, E in the paragraph of "Diffusion model" ? The details of the adopted diffusion model are missing in the main text.

- Experimental evaluation is a bit lacking in several aspects and the results are not fully convincing:
   + It is unclear how the baselines and this method are compared. It is worth noting that the proposed method only generates the foreground object while the other methods also produce the background. Are the background removed before comparison, for the FID computation and user study? It seems unfair if the background is treated differently.
   + A reasonable baseline is to combine amodal segmentation with condition control image generation, which is missing in the comparison.
   + It is unclear how the method is sensitive to the segmentation quality. What if there are different degrees of segmentation error?
   + As mentioned above, it would be more convincing if the method is evaluated on the cases with diverse modes of object shape in the occluded regions.

**Questions:**

See above for detailed questions.

---

> ### Author Response · Authors · 2023-11-15
> **Response to Reviewer 2Pq9 - Part 1**
>
> We thank the reviewer for the time and effort to review our paper. Our answers to the questions are as follows.
>
> ---
>
> **1. The justification for the entire iterative procedure is lacking. It is possible to have noisy output (worse image/inaccurate mask) in both the generation and segmentation stages.**
>
> We would like to provide an analysis of the noise-tolerant capability of the proposed MaskComp. We agree that worse images and inaccurate masks can appear in the generation and segmentation stages respectively. Since all the generation processes are conditioned on the original partial image (as shown in Figure 1), both worse images and inaccurate masks will just influence the conditioned mask for the next iteration. We have two mechanisms to mitigate the influence of the noisy conditioned masks:
>
> - *Error Suppression through Mask Voting*: We vote among multiple masks to form the input of the next iteration. In this way, if only a few masks are degraded, the impact of them can be recovered by the voting operation.
>
> - *Error Tolerance in IMD Iteration*: We train the mask-denoising ControlNet to handle a wide range of occluded masks. Consequently, if the conditioned mask undergoes minimal improvement or degradation due to the noises in a given iteration, it can still be improved in the subsequent iteration. While this may slightly extend the convergence time, it is not anticipated to have a significant impact on the ultimate image quality.
>
> In this way, we consider the IMD process can work in a robust manner. We will show a quantitative analysis in the later response to question 7.
>
> ---
>
> **2. The proposed method lacks mode diversity for object completion due to the averaging operator in the IMD process.**
>
> We agree that MaskComp lacks shape diversity due to the averaging operator. However, we consider that object completion tasks typically only prioritize the estimation closest to the ground truth object. For example, in the closely related amodal segmentation task, the evaluation is conducted by calculating the IoU between the predicted mask and the GT mask. We consider retaining only the most plausible estimation to be reasonable since the rough shape of objects can be deduced by the visible parts in most cases.
>
> In addition, we consider the lack of shape diversity will not hinder the usage of MaskComp in practice. We discuss the potential application of MaskComp in Figure 10. Since the current image editing models struggle to modify partial images, MaskComp can serve as the first step to complete objects and then feed to an image editing model to further process the image. We show another cool application, layered image generation, of MaskComp as shown in Figure A ([github.com/iclr23anonymous739/application.pdf](https://github.com/iclr23anonymous739/rebuttal/blob/main/application.pdf)).
>
> ---
>
> **3. In the generation stage, how does the interpolation is implemented? How does the method generate the interpolated masks and its time embedding?**
>
> We designed several occlusion strategies to create diverse occlusion (the details can be found in Line 417-424). During training, we conduct the random occlusion process twice for each complete mask $M_c$. The partial mask $M_p$ is achieved by considering the occluded areas in both of the occlusion processes. And the interpolated mask $M$ is generated by using one of the occlusions. The time embedding used for the gating operation is shared with the time embedding for encoding the diffusion step in the stable diffusion.
>
> ---
>
> **4. What are the notations M, E in the paragraph of "Diffusion model"? The details of the adopted diffusion model are missing in the main text.**
>
> The $M$ denotes the interpolated mask (Line 146) with an occlusion rate between the complete mask $M_c$ and partial mask $M_p$ and $E$ denotes the text prompt of the object (please note that the text prompt is necessary to fine-tune stable diffusion while optional during inference). We leverage frozen stable diffusion (Line 242) as our underlining diffusion model. We further clarified the notations in the revised version.
>
> ---
>
> **5. It is unclear how the baselines and this method are compared. Are the background removed before comparison, for the FID computation and user study?**
>
> Yes, we aim to ignore the background during all the evaluations. For the quantitative comparison, the computing of FID only considers the ground-truth object area to eliminate the influence of the generated background (Caption of Table 1). For the user study, we give detailed instructions to participants to focus only on the foreground object region and ignore the difference presented in the background (Line 457).

---

> ### Author Response · Authors · 2023-11-15
> **Response to Reviewer 2Pq9 - Part 2**
>
> **6. Comparison with a combination of amodal segmentation and condition image generation as a baseline.**
>
> |Method|FID|
> | ----| :----: |
> |AISFormer+ControlNet| 29.4 |
> |MaskComp| 16.9 |
>
> Table A: Performance comparison with amodal segmentation baseline.
>
> Thanks for the suggestion. We conducted an additional experiment with the masks from the SOTA open-sourced amodal segmentation method ([A] AISFormer) and leveraged ControlNet to generate images based on the amodal masks. We report the performance as shown in Table A. We notice that our method shows obvious superior performance compared to the baseline setting.
>
> [A] AISFormer: Amodal Instance Segmentation with Transformer
>
> ---
>
> **7. What if there are different degrees of segmentation error?**
>
> |Noise degree|Iter 1|Iter 3|Iter 5|Iter 7|Iter 9|
> | ----| :----: | :----: | :----: | :----: | :----: |
> |15\% area| 28.4 | 22.7| 18.9 | 17.2 |16.5|
> |10\% area| 26.4 | 21.4 | 18.1 | 17.0| 16.4|
> |5\% area| 24.9 | 19.6 | 17.0 | 16.2 | 16.0|
> |No noise| 24.7 | 19.4 | 16.9 | 16.1 | 15.9|
>
> Table B: Ablation with noisy conditioned mask. We leverage FID to evaluate the performance. The error is added based on the size of the foreground object.
>
> We conducted an experiment to manually add random errors to the masks. As shown in Table B, we ablate on the iteration number and the degree of segmentation error. We notice that the segmentation error will just increase the converge iteration number while the final performance will not be severely influenced. In addition, since MaskComp predicts a black background, it is easy for segmentation models to segment the foreground objects. Therefore, a large error in segmentation is not expected.
>
> ---
>
> **8. It would be more convincing if the method is evaluated on the cases with diverse modes of object shape in the occluded regions.**
>
> |Occ.|20\%|40\%|60\%|80\%|
> | ----| :----: | :----: | :----: | :----: |
> |FID| 13.4 | 15.7 | 17.2 | 29.9 |
>
> Table 2 (b): Performance with different occlusion rates.
>
> |Occ.|Rectangle|Oval|Object|
> | ----| :----: | :----: | :----: |
> |FID| 15.3 | 15.1| 16.9 |
>
> Table C: Performance with different occlusion types. Object denotes the occlusion is created with a random object shape.
>
> We reported the performance with different occlusion rates as shown in Table 2 (b). We additionally report the performance with different occlusion types as shown in Table C. Since the object occlusion has a more complex boundary, the results with rectangle and oval occlusions show a lower FID than that with more complex object occlusions.

---

> ### Author Response · Authors · 2023-11-22
> **Followup Response to Reviewer 2Pq9**
>
> Thanks for your time and effort in reviewing our work. We hope our earlier response can resolve your concern. As the deadline for the discussion period is approaching, if any aspect of our response or the method itself remains unclear, please do not hesitate to reach out. We are more than willing to provide further clarification.
>
> Here is a summary about the suggested changes to the manuscript:
> - Further discussion about the errors and robustness of the IMD process (Line 317-337).
> - Further explanation about the interpolated mask and time embedding (Line 253-257).
> - Clarification of the $M$, $E$ and the underlined diffusion model (Line 153-158).
> - Comparison with a combination of amodal segmentation and condition image generation (Table 4 (c) & Line 354-359).
> - Performance with different occlusion types (Table 4 (d) & Line 359-360).
>
> We look forward to your continued feedback and sincerely hope that you can improve the rating based on the responses.

---

> ### Comment · Reviewer_2Pq9 · 2023-11-23
> **Thanks for your response**
>
> Thanks for the detailed response.
>
> - 1&7. The argument on the robustness does not fully convince me. The voting does help but it would hurt diversity, which is a dilemma. While enlarging the range of masks may help, there may also be cases which the training does not cover. Thanks for the empirical study, which does show the method is stable on this benchmark.
>
> - 2. The groundtruth may not be unique, especially for articulated objects.
>
> - 3-5 Thanks for the clarification. It would be better to apply a GT mask before sending the results to any evaluation process.
>
> - 6. Thanks for the additional results, which addressed my previous concern.
>
> - 8. Thanks for the results, but I would like to clarify that "diverse modes of object shape" refer to potential multiple GT masks due to different poses of complex objects, such as human/animal classes.

---

> > ### Author Response · Authors · 2023-11-23
> > **Response to Reviewer 2Pq9**
> >
> > We thank the reviewer for the feedback. We would like to provide a further explanation of the remaining weaknesses 2 & 8.
> >
> > ---
> > **Weakness 2 (lack of **shape** diversity): "The groundtruth may not be unique, especially for articulated objects."**
> >
> > We acknowledge that there can be multiple possible GT complete objects. However, we would like to highlight that:
> > - The object's shape is highly constrained by the visible part of the object, which can make the most of the potential GTs to be similar. **The amodal COCO dataset [B] conducted a study among human annotators and demonstrated that the consistency among the annotated GT masks is very high.** We consider most of the objects to be articulated in the amodal COCO. We show a visualization of the masks from multiple human annotators in Figure B ([github.com/iclr23anonymous739/amodal_consistency.png](https://github.com/iclr23anonymous739/rebuttal/blob/main/amodal_consistency.png)). Therefore, we consider retaining only the most plausible estimation to be reasonable since the rough shape of objects can be deduced by the visible parts in most cases.
> > - In addition, we would like to highlight that the mask voting operation just reduces the shape diversity while not influencing the color diversity in nature.
> > - As analyzed in the initial response, we consider the lack of shape diversity will not hinder the usage of MaskComp in practice.
> >
> > [B] Semantic Amodal Segmentation, CVPR 2017
> >
> > ---
> >
> > **Weakness 8 (comparison with different object modes in the occluded area): I would like to clarify that "diverse modes of object shape" refer to potential multiple GT masks due to different poses of complex objects, such as human/animal classes.**
> >
> > We thank the reviewer for the clarification. Since only a few hours remain before the end of the rebuttal, we are regrettably unable to provide the results. We promise to conduct the experiments and add the results in the revised version.
> >
> > We provided ablation about the occlusion rate and occlusion type. We hope the ablation of those aspects can also help understand the property of the proposed method.
> >
> > ---
> >
> > We look forward to your continued feedback and sincerely hope that the responses can clarify the remaining concerns.

---

> > > ### Comment · Reviewer_2Pq9 · 2023-11-23
> > > **Further comments**
> > >
> > > Thanks for the summary. In my opinion, it is important to cope with uncertainty in such completion problems, and this challange needs to be addressed properly in the first place.

---

> ### Author Response · Authors · 2023-11-23
> **Thanks for the response**
>
> We thank the reviewer for the feedback. However, we do not agree with the reviewer due to the lack of concrete reasons to support his point. We consider we have explained the unnecessity of shape diversity in the completion task from the (1) conventional (amodal evaluation),  (2) practical (application) and (3) empirical (analysis of human annotator) perspectives and have provided convincing evidence to support our claim that sacrificing shape diversity to gain reliability and image quality is deserving.
>
> We would encourage the reviewer to discuss the specific concern with other reviewers and AC during the discussion phase. Thank you.

---

### Official Review · Reviewer_PKfR · 2023-11-01

**Soundness:** 3 good
**Presentation:** 3 good
**Contribution:** 2 fair
**Rating:** 6
**Confidence:** 5

**Summary:**

This paper aims to generate a complete object given partial observation. Instead of designing an end-to-end pipeline, this paper introduces object masks as an intermediate representation to complete the object interactively. Specifically, in each iteration, the proposed method generates a set of completions using stable diffusion and extracts the corresponding masks with a segmentation model. These masks are fused and fed into the generation process for the next iteration. Experiments and analysis demonstrate the effectiveness of the proposed method.

**Strengths:**

1. The proposed method (MaskComp) introduces a novel interactive approach to complete an object by generating object masks as guidance.

1. Section 3.3 tries to give some theoretical analysis of MaskComp, which is interesting and helps to understand the benefit of introducing masks in the generation approach.

1. From the visual results, I find MaskComp completes the input partial object and achieves higher perceptual quality compared to other methods. Quantitatively, it also achieves higher FID and user study scores.

1. The paper's presentation is good, making it easy to follow and understand.

**Weaknesses:**

1. The technical contributions of the proposed method could be further improved. For now, the proposed method is mainly a mask-guided stable diffusion model with an off-the-shelf segmentation model to produce the mask condition. Using SAM to generate masks is straightforward and the mask voting process gives no surprises.

1. If generating a mask is the key to generating high-quality images, why not directly use an encoder-decoder model like U-Net or an SD to predict the target complete mask in one step? As generating the mask is relatively a simple task, I believe a U-Net might be enough to obtain an accurate object mask (this can be regarded as an outpainting task for binary images). It would be interesting to have conducted such an experiment.

1. More ablations study of the proposed method should be given. E.g., different segmentation models, different mask voting strategies,

1. One of the main drawbacks of the proposed method is its slow inference speed. Each iteration in the generation process involves generating multiple image candidates and their segmentations (segment anything model is quite slow), let alone running for multiple iterations. The authors did not report the running time statistics of different methods and omitted this limitation in the limitations section. I encourage the authors to have more comparisons and discussions of the running time.

1. The FID metric shown in Table 1 is problematic. The FID only considers the ground-truth object area; however as shown in Figure 6, different methods generate different foreground areas that may not match the ground-truth object mask. Thus, this metric will yield a higher FID score if the generated object has a large discrepancy with the ground truth even if it is realistic.

1. I cannot find the qualitative comparisons with ControlNet and Kandinsky.

1. I would like to see the results of using the ground-truth mask as input to see how different methods perform with such an oracle. This also tells the generation upper bound of mask-guided methods.

**Questions:**

The authors should provide more experiments and discussions in the rebuttal to address the weaknesses raised above.

---

> ### Author Response · Authors · 2023-11-15
> **Response to Reviewer PKfR - Part 1**
>
> We thank the reviewer for the time and effort to review our paper. Our answers to the questions are as follows.
>
> ---
>
> **1 & 2. Using SAM to generate masks is straightforward and the mask voting process gives no surprises. If generating a mask is the key to generating high-quality images, why not directly use an encoder-decoder model like U-Net or an SD to predict the target complete mask in one step?**
>
> |Method|SD|UNet |IMD (Ours)|
> | ----| :----: |:----: |:----: |
> |IoU| 82.4 | 75.4 |88.5 |
>
> Table A: IoU of predicted masks on AHP dataset.
>
> In this paper, we investigate the relation between object generation and segmentation. We found that even if the segmentation model is not trained, the segmentation can still benefit the generation. It is a good idea to conduct experiments to investigate other segmentation models.
> We report the results of using a ControlNet (frozen SD) and UNet with the partial object as the condition to predict the complete mask in Table A. We notice that the complete mask predicted from the IMD process shows obviously higher IoU to the ground truth which indicates the IMD process is an effective mask prediction approach.
>
> In addition, we would like to clarify that the mask prediction is not simply conducted by the SAM but by the entire IMD process with both generation and segmentation stages. We treat the segmentation as the shape property of the generated object. Therefore, with an object generation model, we can directly obtain the shape using SAM without training another mask-specific model.
>
> We also want to further clarify the usage of the voting operation. Given a partial object $I_p$, the generation model samples from a distribution that contains both realistic and unrealistic images. As shown in Figure A ([github.com/iclr23anonymous739/dist.pdf](https://github.com/iclr23anonymous739/rebuttal/blob/main/dist.pdf)), since the image distribution is complex, the expectation $E[I_c]$ cannot represent a realistic image. However, when we consider the shape of the generated images, we are excited to find that the expectation $E[\mathcal{S}(I_c)]$ leads to a more realistic shape. We consider this observation interesting as, for most of the other generation tasks (non-conditioned), averaging the object shape will just yield an unrealistic random shape. This observation (Figure 4) serves as one of our core observations to build the IMD process. Here, SAM serves as the tool to extract the object shape and voting is a way to binarize the expectation of object shapes to a binary mask.
>
> ---
>
> **3. More ablation studies of the proposed method should be given. E.g., different segmentation models, and different mask voting strategies.**
>
> |Method|Mask2Former|ClipSeg|SAM|
> | :----| :----: | :----: | :----: |
> |FID| 22.5| 19.9 |16.9|
>
> Table B: Performance with different segmentation models.
>
> |Method|Voting with logits|Mean with logits|Voting with mask| Mean with mask|
> | :----| :----: | :----: | :----: | :----: |
> |FID| 16.9| 17.2 |17.6| 17.0 |
>
> Table C: Performance with different voting strategies. Logits: mask logits before binarzing.
>
> We conducted ablation experiments to determine the design choice in the segmentation stage. We report the ablation studies about segmentation models and voting strategies in Table B and Table C. The current design choice of using SAM and voting with logits is based on the ablation results.
>
> ---
>
> **4. Comparisons and discussions of the running time.**
>
> |Reduce SD step|Generation stage|Segmentation Stage|Total|FID|
> | :----:| :----: | :----: | :----: | :----: |
> || 14.3s | 1.2s |15.5s |16.9|
> |$\checkmark$| 8.6s | 1.2s|9.8s| 17.4|
>
> Table D: Inference time for each stage of IMD on a single V100 GPU.
>
> Thank you for your suggestion. We demonstrate the running time of each component in IMD as shown in Table D. We acknowledge that the inference speed of MaskComp is slow. To improve the inference speed, we notice that decreasing the diffusion step number in the first several IMD steps will not severely degrade the performance. By incorporating this idea into MaskComp, the average running time was reduced to 2/3 original time. We will add the discussion to our limitation section.
>
> ---
>
> **5. The FID only considers the ground-truth object area. Thus, this metric will yield a higher FID score if the generated object has a large discrepancy with the ground truth even if it is realistic.**
>
> |Method|ControlNet|Kandinsky 2.1|SD 1.5|SD 2.1|MaskComp (Ours)|
> | ----| :----: | :----: | :----: | :----: | :----: |
> |AHP|45.4|43.9|41.4|39.9|21.3|
> |DYCE|49.4|47.7|43.4|41.1|25.4|
>
> Table E: Performance comparison with FID scores calculated with SAM masks.
>
> Thanks for the comment. We report the FID scores evaluated by cropping the object with SAM in Table E. We notice that, for the FID scores calculated with both GT mask and SAM mask, our method outperforms baseline methods with an obvious marginal which indicates the effectiveness of the proposed method.

---

> ### Author Response · Authors · 2023-11-15
> **Response to Reviewer PKfR - Part 2**
>
> **6. Qualitative comparisons with ControlNet and Kandinsky.**
>
> Thanks for pointing this out. We add additional comparisons with ControlNet and Kandinsky in Figure 6 ([github.com/iclr23anonymous739/comparison.pdf](https://github.com/iclr23anonymous739/rebuttal/blob/main/comparison.pdf)). We notice that MaskComp achieves superior performance compared to baseline methods.
>
> ---
>
> **7. The results of using the ground-truth mask as input to see how different methods perform with such an oracle (This tells the generation upper bound of mask-guided methods).**
>
> |Mask|Visible|Noisy|Complete|
> | ----| :----: | :----: | :----: |
> |FID| 16.9| 15.3 |12.7|
>
> Table 2 (a): Performance with different conditioned masks. Occlusion rate: partial mask > noisy mask > complete mask.
>
> We reported the results of using the ground-truth mask as the condition in Table 2 (a) and there is also a visual comparison among different conditioned masks in Figure 2. With the ground-truth mask, the generation model shows a promising performance to complete the objects. This also serves as the core motivation for our iterative mask denoising process (IMD) which starts from a partial mask to gradually generate a complete mask to boost the generation process.

---

> ### Author Response · Authors · 2023-11-22
> **Followup Response to Reviewer PKfR**
>
> We deeply appreciate the time and effort invested in reviewing our work and providing valuable feedback. Since the deadline for the discussion period is approaching and we haven't heard back from you, here's a summary of our earlier response to the concerns raised:
> - *Weakness 1 & 2*: We discuss the motivation for using of SAM and voting with our observations and conduct additional experiments to compare the IMD process with baselines. Moreover, we clarify that the mask prediction is not simply conducted by the SAM but by the entire IMD process with both generation and segmentation stages.
>
> - *Weakness 3*: We provide the suggested ablation results and incorporate the discussion in Table 4 & Line 349-353 in the revised version.
>
> - *Weakness 4*: We discuss the inference speed and incorporate the analysis in Table 3 (c) and Line 295-300 in the revised version.
>
> - *Weakness 5*: We report the FID score calculated with masks predicted from the generated images and update the Table 1 to better illustrate the performance comparison.
>
> - *Weakness 6*: We update Figure 6 to additionally compare with ControlNet and Kandinsky.
>
> - *Weakness 7*: We further discuss the results in Table 3 (a) about the performance with GT mask as condition.
>
> We are open to further discussions if there are any aspects that remain unclear. We look forward to your continued feedback and sincerely hope that you can improve the rating based on the responses.

---

### Official Review · Reviewer_xAwN · 2023-11-02

**Soundness:** 3 good
**Presentation:** 3 good
**Contribution:** 2 fair
**Rating:** 6
**Confidence:** 3

**Summary:**

This paper presents a method to complete an object which undergoes occlusion. The algorithm alternats generation and segmentation stages to infer the shape and texture of the original object without occlusion. The segmentation and generation helps each other to refine the  result as the iteration goes on. Diffusion model generates the image utilizing the mask info as a condition. Segmentation is basically derived from the generation result; for better result multiple instance of images are generated and their segments are averaged to yield segmentation mask. The experimental result shows IFD comparision with some recent researches, and human assessment is also performed to compare the results.

**Strengths:**

Clever idea improved the result of occluded object completion effectively. The recent progress of image generation models are actively analyzed and the authors found useful problem task.
In the quantitative evaluation the FID metric shows significant performance compared other method, and the numbers are convinced by showing qualitative results.
Greatly overcame the random unstable results which occurs frequently from the image generation model by averaging the results of multiple runs.

**Weaknesses:**

While this paper has attractive strengths, this research is rather applicational research that exploits good features of prior researches. Considering the overall direction of the papers presented in this conference (ICLR), readers may expect more theoretical idea or fundamental thas can be transferrable to of stimulate other research. This paper is heavely dependent on Zhang 2023 paper.

**Questions:**

The user study is included in this research to evaluate the quality of object completion.
If more details of the user study are provided, it will be more convincing. In many other research areas user study is conducted; and to dispel any latent bias or mistakes they generally offer user study method and protocol, such as: how many persons participate? how to select the subjects? what exactly were the sentences for questions? how was the user interface or the testing environment?

---

> ### Author Response · Authors · 2023-11-15
> **Response to Reviewer xAwN**
>
> We thank the reviewer for the time and effort to review our paper. Our answers to the questions are as follows.
>
> ---
>
> **1. Readers of ICLR may expect more theoretical ideas or fundamentals that can be transferrable to stimulate other research. This paper is heavily dependent on Zhang 2023 paper.**
>
> Thanks for the important comment. We would like to further clarify that the idea behind MaskComp is to investigate the relation between image generation and segmentation. Previous work PaintSeg [A] has proved that image generation can benefit the segmentation and lead to training-free object segmentation capability. In this work, our results indicate that segmentation can also be beneficial to image generation with the proposed IMD process. To better understand the process, we also provide a theoretical analysis of the process in Section 3.3. We consider that the idea of leveraging segmentation to boost generation can be of interest to the community.
>
> [A] PaintSeg: Training-free Segmentation via Painting, NeurIPS 2023
>
> ---
>
> **2. If more details of the user study are provided, it will be more convincing. How many people participate? How to select the subjects? What exactly were the sentences for questions? How was the user interface or the testing environment?**
>
> We provide information about the user study in the supplementary materials (Line 450-464). There are 16 people participated in the user study. All participants were selected from graduate students who have relevant knowledge to understand the task. We illustrate the instructions for the participants in Line 456-564. We demonstrate the user interface in Figure A ([github.com/iclr23anonymous739/platform.png](https://github.com/iclr23anonymous739/rebuttal/blob/main/platform.png)).

---

> > ### Comment · Reviewer_xAwN · 2023-11-22
> >
> > Thanks to the authors. Although my final overall rate will not be changed, the user study parts are clarified by the comment and spplementary material.

---

> > > ### Author Response · Authors · 2023-11-22
> > > **Thank you for the valuable feedback**
> > >
> > > We sincerely thank the reviewer for the valuable feedbacks to improve our submission!

---

### Comment · Area_Chair_uWrc · 2023-11-17
**Please engage in reviewer-author discussion**

Dear reviewers,

The paper got diverging scores. The authors have provided their response to the comments. Could you look through the other reviews and engage into the discussion with authors? See if their response changes your assessment of the submission?

Thanks!
AC

---

### Author Response · Authors · 2023-11-23
**Rebuttal Summary**

We thank the reviewers and AC for the time and effort in handling our submission. We appreciate that all four reviewers have a consensus that our work is novel and interesting. To facilitate the decision making, we would like to give a rebuttal summary:

---

**Reviewer xAwN**

We addressed most of the concerns and the reviewer would like to keep the positive rating.

---

**Reviewer t7dr**

All concerns (initial and after reading other reviewer's comments) are addressed.

---

**Reviewer 2Pq9**

We have addressed most of the concerns and we summarize the major remaining concern here.

>Weakness 2 (lack of **shape** diversity): The proposed method lacks mode diversity for object completion due to the averaging operator in the IMD process.

We consider the necessity of shape diversity in the conditional object completion task as the major concern remaining. We do not agree with it since the reviewer failed to provide concrete reasons to support his point after the rebuttal. We provided the analysis from the (1) conventional (amodal evaluation), (2) practical (application), and (3) empirical (analysis of human annotator) perspectives. We consider that the provided evidence can support our claim that sacrificing shape diversity to gain reliability and image quality in the object completion task is deserving.
- **We consider that object completion tasks typically only prioritize the estimation closest to the ground truth object, e.g. amodal segmentation.** For example, in the closely related amodal segmentation task, the evaluation is conducted by calculating the IoU between the predicted mask and the GT mask. We consider retaining only the most plausible estimation to be reasonable since the rough shape of objects can be deduced by the visible parts in most cases.
- **We consider the lack of shape diversity will not hinder the usage of MaskComp in practice.** We discuss the potential application of MaskComp in Figure 10. Since the current image editing models struggle to modify partial images, MaskComp can serve as the first step to complete objects and then feed to an image editing model to further process the image. We show another cool application, layered image generation, of MaskComp as shown in Figure A ([github.com/iclr23anonymous739/application.pdf](https://github.com/iclr23anonymous739/rebuttal/blob/main/application.pdf)).

The reviewer was further concerned that the **"GT may not be unique, especially for articulated objects."** We acknowledge that there can be multiple possible GT complete objects. However, we would like to highlight that:
- The object's shape is highly constrained by the visible part of the object, which can make the most of the potential GTs to be similar. **The amodal COCO dataset [B] conducted a study among human annotators and demonstrated that the consistency among the GT masks is very high**. We consider most of the objects to be articulated in the amodal COCO. We show a visualization of the masks from multiple human annotators in Figure B ([github.com/iclr23anonymous739/amodal_consistency.png](https://github.com/iclr23anonymous739/rebuttal/blob/main/amodal_consistency.png)).
- In addition, we would like to highlight that the mask voting operation just reduces the shape diversity while not influencing the color diversity in nature. And as mentioned by Reviewer xAwN, the proposed method can "Greatly overcome the random unstable results which occur frequently from the image generation model by averaging the results of multiple runs."

**Considering all the factors at play, we consider sacrificing shape diversity to gain reliability and image quality in the object completion task is deserving.**

[B] Amodal Semantic Segmentation, CVPR 2017

---

**Reviewer PKfR**

Though Reviewer PKfR does not provide further feedback to our response, we consider the detailed discussion and comprehensive experiments can address his concerns effectively. Here's a summary of our response to the concerns raised:

- *Weakness 1 & 2*: We discuss the motivation for using of SAM and voting with our observations and conduct additional experiments to compare the IMD process with baselines. Moreover, we clarify that the mask prediction is not simply conducted by the SAM but by the entire IMD process with both generation and segmentation stages.

- *Weakness 3*: We provide the suggested ablation results and incorporate the discussion in Table 4 & Line 349-353 in the revised version.

- *Weakness 4*: We discuss the inference speed and incorporate the analysis in Table 3 (c) and Line 295-300 in the revised version.

- *Weakness 5*: We report the FID score calculated with masks predicted from the generated images and update Table 1 to better illustrate the performance comparison.

- *Weakness 6*: We update Figure 6 to additionally compare with ControlNet and Kandinsky.

- *Weakness 7*: We further discuss the results in Table 3 (a) about the performance with GT mask as the condition.

---

### Meta-Review · Area_Chair_uWrc · 2023-12-06

**Metareview:**

This paper aims at boosting the object completion by alternating between a mask-conditioned image generation stage and a segmentation stage. In the generation stage a conditional diffusion U-Net is trained to generate a complete object; in the segmentation stage, a more accurate mask is obtained by leveraging segmentation result of the generated object image. The proposed method is evaluated on AHP and DYCE datasets with comparisons to several diffusion-based baselines.

Strengths
This work is well motivated and the proposed IDM process with alternative generation and segmentation has certain novelty. Experimental results demonstrate the effectiveness of the proposed method.

Weaknesses
The balance between robustness and diversity is not well explained and justified. Lack comparison with other methods in terms of amodal segmentation performance.

**Justification For Why Not Higher Score:**

In general, the effectiveness about the proposed IDM process still needs to be well justified. Especially the balance between its robustness and diversity.

**Justification For Why Not Lower Score:**

The proposed object completion pipeline with alternative generation and segmentation has certain novelty. Experimental results demonstrate the effectiveness of the proposed method.

---

### Decision · Program_Chairs · 2024-01-16

Reject